# Genome-wide association and transcriptome studies identify target genes and risk loci for breast cancer

Manuel A. Ferreira et al.[#]

Genome-wide association studies (GWAS) have identified more than 170 breast cancer susceptibility loci. Here we hypothesize that some risk-associated variants might act in non-breast tissues, specifically adipose tissue and immune cells from blood and spleen. Using expression quantitative trait loci (eQTL) reported in these tissues, we identify 26 previously unreported, likely target genes of overall breast cancer risk variants, and 17 for estrogen receptor (ER)-negative breast cancer, several with a known immune function. We determine the directional effect of gene expression on disease risk measured based on single and multiple eQTL. In addition, using a gene-based test of association that considers eQTL from multiple tissues, we identify seven (and four) regions with variants associated with overall (and ER-negative) breast cancer risk, which were not reported in previous GWAS. Further investigation of the function of the implicated genes in breast and immune cells may provide insights into the etiology of breast cancer.

---

Correspondence and requests for materials should be addressed to M.A.F. (email: Manuel.Ferreira@qimr.edu.au). These authors contributed equally: Jonathan Beesley, Georgia Chenevix-rench. [#]A full list of authors and their affiliations appears at the end of the paper.

Breast cancer is the most commonly diagnosed malignancy and most frequent cause of cancer-related mortality in women[1]. Genome-wide association studies (GWAS) have detected more than 170 genomic loci harboring common variants associated with breast cancer risk, including 20 primarily associated with risk of ER-negative disease[2,3]. Together, these common variants account for 18% of the two-fold familial relative risk of breast cancer[2].

To translate GWAS findings into an improved understanding of the biology underlying disease risk, it is essential to first identify the target genes of risk-associated variants. This is not straightforward because most risk variants lie in non-coding regions, particularly enhancers, many of which do not target the nearest gene[4]. To help with this task, we recently developed a pipeline that identifies likely target genes of breast cancer risk variants based on breast tissue-specific genomic data, such as promoter–enhancer chromatin interactions and expression quantitative trait loci (eQTL)[2]. Using this approach, called INQUISIT, we identified 689 genes as potential targets of the breast cancer risk variants. However, it is likely that at least some breast cancer risk variants modulate gene expression in tissues other than breast, which were not considered by INQUISIT; for example, breast cancer risk variants are enriched in histone marks measured in adipose tissue[2]. On the other hand, the immune system also plays a role in the elimination of cancer cells[5] so it is possible that some breast cancer risk variants influence the expression of genes that function in the immune system.

The first aim of this study was to identify additional likely target genes of the breast cancer risk variants identified by the Breast Cancer Association Consortium[2,3] using information on eQTL in multiple relevant tissue types: adipose, breast, immune cells, spleen, and whole-blood. The second aim was to identify previously unreported risk loci for breast cancer by formally integrating eQTL information across tissues with results from the GWAS[2,3] using EUGENE, a recently described gene-based test of association[6,7], that is conceptually similar to other transcriptome-wide association study (TWAS) approaches, such as PrediXcan[8]. Gene-based analyses would be expected to identify previously unreported risk loci if, for example, multiple independent eQTL for a given gene are individually associated with disease risk, but not at the genome-wide significance level used for single-variant analyses.

## Results

**Predicted target genes of overall breast cancer risk variants.** Using approximate joint association analysis implemented in GCTA[9] (see Methods), we first identified 212 variants that were independently associated (i.e. with GCTA-COJO joint analysis $P < 5 \times 10^{-8}$) with breast cancer in a GWAS dataset of 122,977 cases and 105,974 controls[2] (Supplementary Data 1). Of note, 20 of these variants reached genome-wide significance in the joint, but not in the original single-variant, association analysis; that is, they represent secondary signals that were masked by the association with other nearby risk variants, as described previously[9].

We extracted association summary statistics from 117 published eQTL datasets identified in five broad tissue types: adipose, breast, individual immune cell types, spleen and whole-blood (Supplementary Data 2). For each gene and for a given eQTL dataset, we identified *cis* eQTL (within 1 Mb of gene boundaries) in low linkage disequilibrium (LD; $r^2 < 0.05$) with each other, and with an association with gene expression significant at a conservative significance threshold of $8.9 \times 10^{-10}$. We refer to these as "sentinel eQTL". The mean number of sentinel eQTL per gene ranged from 1.0 to 2.9 across the 117 eQTL datasets considered, which varied considerably in sample size and number of genes tested (Supplementary Data 2).

When we intersected the list of variants from the joint association analysis and the list of sentinel eQTL from published datasets, we identified 46 sentinel risk variants that were in high LD ($r^2 > 0.8$) with one or more sentinel eQTL, implicating 88 individual genes at 46 loci as likely targets of breast cancer risk variants (Supplementary Data 3 and 4). Twenty-five risk variants had a single predicted target gene, 10 had two, and 11 had three or more (Supplementary Data 5).

Of the 88 genes, 75 (85%) were identified based on eQTL from whole-blood, 10 (11%) from immune cells (*PEX14, RNF115, TNNT3, EFEMP2, SDHA, AP4B1, BCL2L15, BTN3A2, HIST1H2BL, SYNE1*), and three (4%) exclusively from adipose tissue (*ZNF703, HAPLN4, TM6SF2*) (Supplementary Data 4). Only four sentinel risk variants were in LD with a sentinel eQTL in breast tissue (for *ATG10, PIDD1, RCCD1,* and *APOBEC3B*); all were also eQTL in whole-blood and immune cells. However, it is noteworthy that an additional 29 sentinel eQTL listed in Supplementary Data 3 had a modest, yet significant association with the expression of the respective target gene in breast tissue (GTEx V7, $n = 251$), suggesting that larger eQTL datasets of this tissue will be informative to identify the target genes of sentinel risk variants.

A total of 62 genes were included in the list of 925 targets predicted in the original GWAS using INQUISIT[2], while 26 genes represent previously unreported predictions (Supplementary Data 5). Regional association plots for these 26 genes are presented in Supplementary Fig. 1, with three examples shown in Fig. 1.

**Directional effect of gene expression on breast cancer risk.** For the 88 genes identified as likely targets of breast cancer risk variants, we studied the directional effect of genetically-determined gene expression on disease risk, based on the sentinel eQTL that was in LD with the sentinel risk variant. For each gene, we first determined whether the eQTL allele that was associated with reduced breast cancer risk was associated with higher or lower target gene expression. Of the 77 genes for which this information could be obtained (detailed in Supplementary Data 4), the protective allele was associated with lower expression for 43 genes (e.g. *GATAD2A, FAM175A, KCNN4,* and *CTB-161K23.1*) and higher expression for 28 genes (e.g. *RCCD1, ATG10, ELL,* and *TLR1*) (summarized in Table 1 and Supplementary Data 6). For the remaining six genes (*ADCY3, AMFR, APOBEC3B, CCDC127, HSPA4,* and *MRPS18C*), conflicting directional effects were observed across different tissues, and so the interpretation of results is not straightforward.

**Directional effect based on information from multiple eQTL.** In the previous analysis, the directional effect of gene expression on disease risk was assessed based on a single eQTL at a time. However, the expression of most genes is associated with multiple independent eQTL, which may not have the same directional effect on disease risk. To address this limitation, we assessed if results from the single QTL analyses above were recapitulated by considering information from multiple eQTL using S-PrediXcan[10]. We applied this approach to the same GWAS results[2] and used transcriptome prediction models from whole-blood, generated based on data from the Depression Genes and Network study ($n = 922$[11]) and GTEx ($n = 369$). We used SNP prediction models for gene expression in whole-blood because most genes (75 of 88) were identified as likely targets based on eQTL information from this tissue.

Results from this analysis are presented in Supplementary Data 7. The predicted directional effect of gene expression on disease risk was available in both the single eQTL and

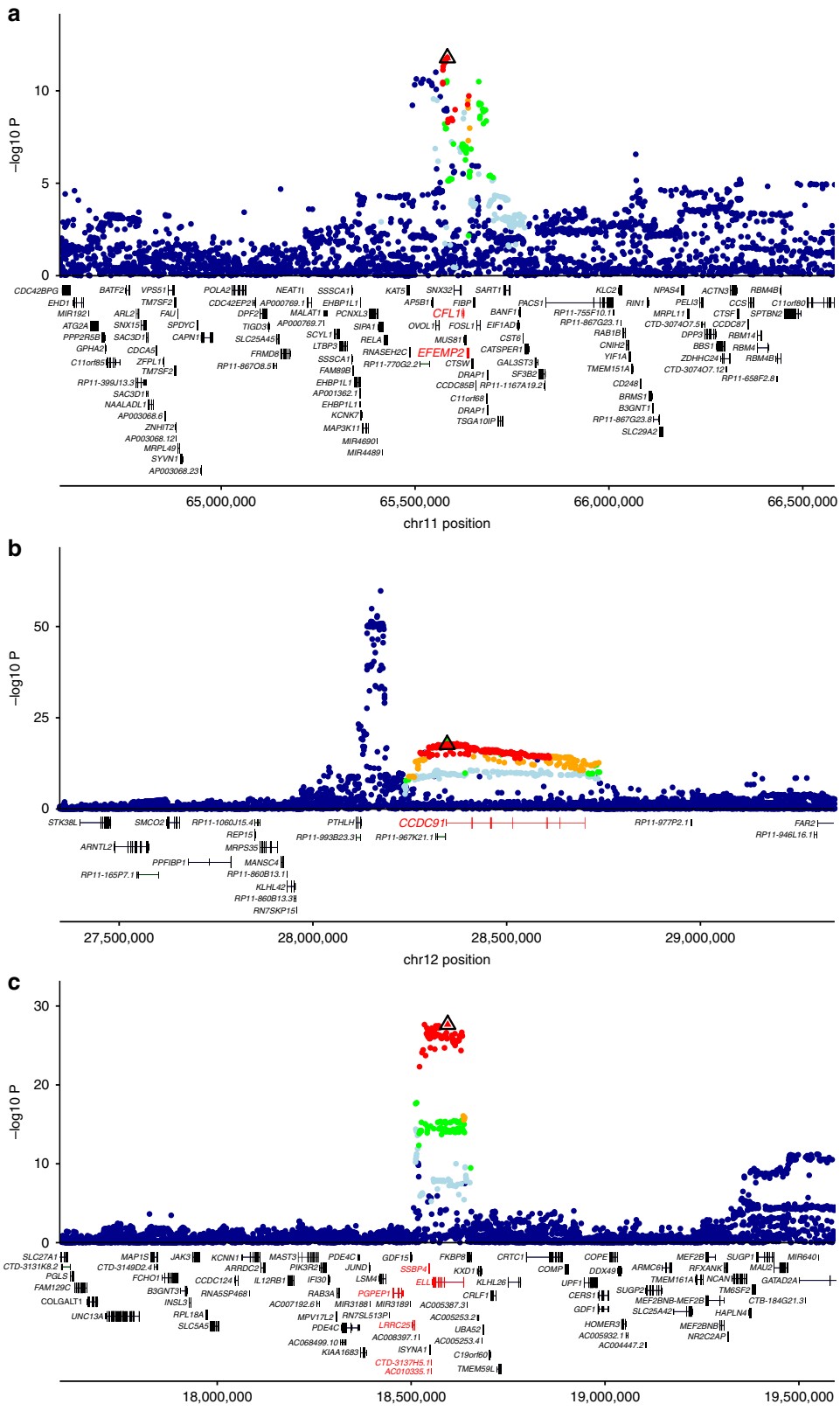

S-PrediXcan analyses for 48 of the 88 likely target genes. For 42 of those 48 genes (88%) the two predictions matched, supporting a consistent directional effect across multiple eQTL of the same gene. The inconsistent results observed for the remaining six genes were likely caused by technical biases (possible explanations in Supplementary Data 7). Similar findings were obtained when considering whole-blood transcriptome prediction models based on data from the GTEx consortium (Supplementary Data 7). Overall, these results indicate very good agreement between the directional effect of gene expression on disease risk obtained using information from individual or multiple eQTL.

**Fig. 1** Examples of previously unreported target gene predictions at known breast cancer risk loci. Variants are represented by points colored according to the LD with the sentinel risk variant (red: ≥0.8, orange: 0.6–0.8, green: 0.4–0.6, light blue: 0.2–0.4, and dark blue: <0.2). Sentinel risk variants (triangles) were identified based on joint association analysis[9]. Figure shows on the y-axis the evidence for breast cancer association ($-\log_{10}$ of the P-value in the original published GWAS results[2], obtained in that study using an inverse-variance meta-analysis), and on the x-axis chromosomal position. Gene structures from GENCODE v19 gene annotations are shown and the predicted target genes shown in red. **a** The sentinel risk variant at this locus (rs875311) was in LD with sentinel eQTL for *CFL1* (in whole blood) and for *EFEMP2* (in CD8[+] T cells only). **b** The sentinel risk variant (rs11049425, target gene: *CCDC91*) represents a secondary association signal in this region. **c** The sentinel risk variant at this locus (rs8105994) is in LD with sentinel eQTL for two previously unreported target gene predictions (*AC010335.1* and *LRRC25*) and four previously predicted targets (*CTD-3137H5.1*, *ELL*, *PGPEP1* and *SSBP4*; (Supplementary Data 5). Regional association plots for the remaining target gene predictions for overall breast cancer (Supplementary Data 3) are provided in Supplementary Figure 1

| Table 1 Directional effect of genetically determined gene expression on disease risk for predicted target genes of breast cancer sentinel risk variants | |
|---|---|
| **Directional effect** | **Predicted target genes of breast cancer sentinel risk variants** |
| Decreased expression associated with decreased risk | *AC007283.5, AHRR, AP006621.5, AP006621.6, APOBEC3B-AS1, ARRDC3, ASCC2, BCL2L15, BTN2A1, CCDC170, CCDC91, CDCA7L, CEND1, CES1, COX11, CTB-161K23.1, CTD-2116F7.1, CYP51A1, DDA1, DFFA, EFEMP2, ENPP7, FAM175A, GATAD2A, HAPLN4, HCG11, HIST1H4L, KCNN4, LRRC25, LRRD1, OGFOD1, PIDD1, PPIL3, PTPN22, RPS23, SIRT5, SMG9, TGFBR2, TM6SF2, TMEM184B, TNS1, ZBTB38, ZNF703* |
| Increased expression associated with decreased risk | *AC010335.1, AKAP9, APOBEC3A, ATF7IP, ATG10, ATP6AP1L, BTN2A3P, CBX6, CENPO, CFL1, COQ5, CTD-3137H5.1, DCLRE1B, DNAJC27, ELL, ESR1, HLF, L3MBTL3, NUDT17, PGPEP1, RCCD1, RHBDD3, RNF115, RP11-486M23.2, SIVA1, SYNE1, TEFM, TLR1* |
| Ambiguous | *ADCY3, AMFR, APOBEC3B, CCDC127, HSPA4, MRPS18C* |

**Target gene predictions supported by functional data**. The 88 genes identified represent target predictions that should be experimentally validated, as outlined previously[12]. To help prioritize genes for functional follow-up, we identified a subset for which publicly available functional data supported the presence of either (i) chromatin interactions between an enhancer and the gene promoter[4,13–15]; or (ii) an association between variation in enhancer epigenetic marks and variation in gene expression levels[16–19]. We only considered enhancers that overlapped a sentinel risk variant (or a proxy with $r^2 > 0.80$) and restricted our analysis to blood cells (Supplementary Data 8), given that most target genes were identified based on eQTL data from whole-blood. We found that 25 (28%) of the 88 target gene predictions were supported by functional data (Supplementary Data 9).

**Previously unreported risk loci for breast cancer**. The second major goal of this study was to identify previously unreported risk loci for breast cancer using gene-based association analyses. We first used approximate conditional analysis implemented in GCTA[9] to adjust the GWAS results[2] (Fig. 2a) for the effects of the 212 variants that had a significant independent association with overall breast cancer. As expected, in the resulting adjusted GWAS no single variant had a genome-wide significant association (i.e. all had a GCTA-COJO conditional association $P > 5 \times 10^{-8}$; Fig. 2b). We then applied the EUGENE gene-based approach[6,7] to the adjusted GWAS results, considering in a single association analysis *cis* eQTL identified in five broad tissue types: adipose, breast, immune cells, spleen, and whole-blood (Supplementary Data 10). That is, we did not perform a separate gene-based analysis for each tissue, but rather a single analysis that considers all eQTL reported across the five tissues.

Of the 19,478 genes tested (full results provided as Supplementary Material), 11 had a significant gene-based association after correcting for multiple testing (EUGENE $P < 0.05/19,478 = 2.5 \times 10^{-6}$; Table 2; Fig. 2c). The specific eQTL included in the gene-based test for each of these 11 genes, which were located in six loci >1 Mb apart, are listed in Supplementary Data 10.

Regional association plots for the 11 genes are presented in Supplementary Fig. 2, with three examples shown in Fig. 3. Except for the *MAN2C1* locus[20], these loci have not previously been identified by GWAS and thus represent putative breast cancer susceptibility loci.

For most (9 of 11) genes identified, the association P-value obtained with the gene-based test was more significant than the P-value obtained with the individual eQTL most associated with disease risk, indicating that multiple sentinel eQTL for the same gene were associated with disease risk (range 2–6 associated eQTL per gene; Table 2). For example, the EUGENE gene-based P-value for *GSTM2* was $6.6 \times 10^{-8}$, while the best individual eQTL showed more moderate association with breast cancer risk (GCTA-COJO conditional association $P = 4.1 \times 10^{-5}$; five of the 14 sentinel eQTL tested for this gene were nominally associated with disease risk (Supplementary Data 11).

We also studied the predicted directional effect of gene expression on disease risk, as described above for target genes of known breast cancer risk variants. When we considered information from all eQTL associated with disease risk for each of the 11 genes (Supplementary Data 11), we found that decreased disease risk was consistently associated with decreased gene expression for three genes and increased expression for five genes (Table 3 and Supplementary Data 12). For the remaining three genes, inconsistent directional effects were observed across different eQTL.

Lastly, we used EUGENE to determine if any of the 88 target genes of sentinel risk variants identified based on individual eQTL also had a significant gene-based association in the adjusted GWAS results. This would indicate that information from additional breast cancer risk variants (i.e. in low LD with the sentinel risk variants) supported the original target gene prediction, which could be used to prioritize genes for functional follow-up. We found that 11 of the 88 target genes had a nominally significant gene-based association in the adjusted GWAS results (EUGENE $P < 0.05$; Supplementary Data 13), with one remaining significant after correcting for multiple testing: *CBX6* (EUGENE $P = 0.0002$).

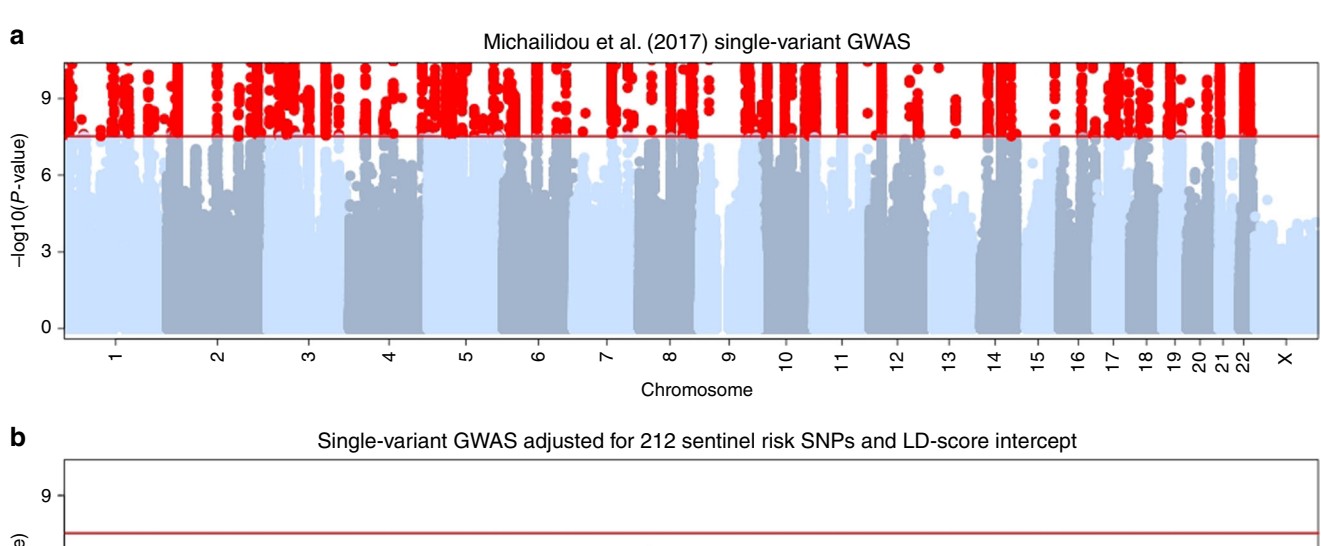

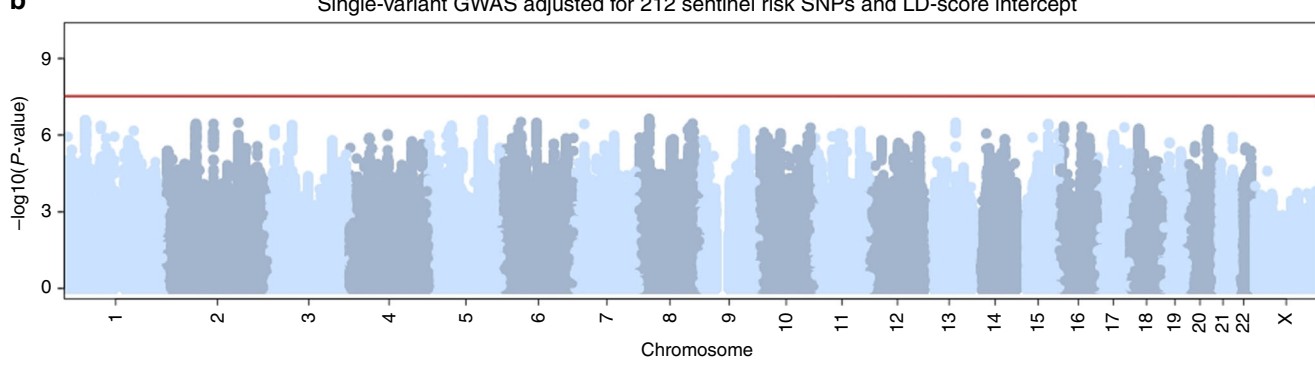

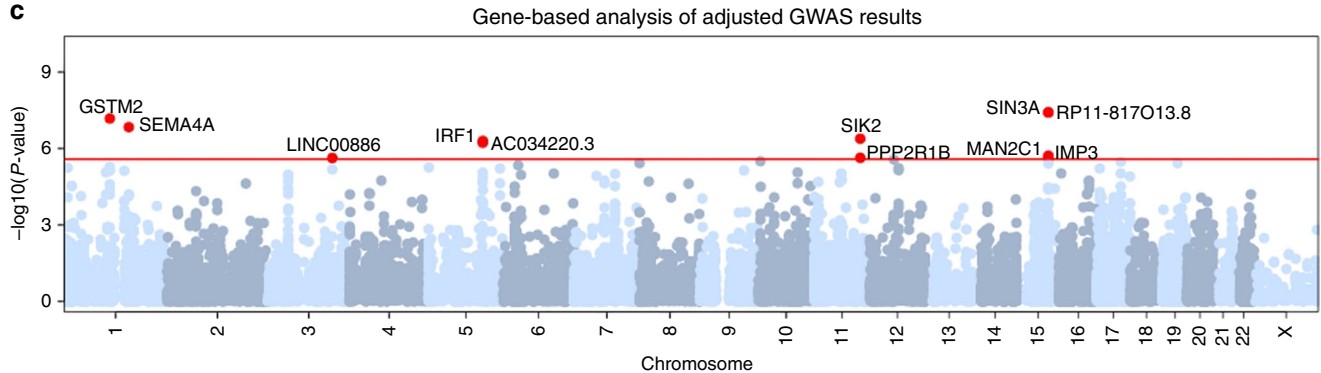

**Fig. 2** Manhattan plots summarizing association results for overall breast cancer. **a** Association results ($-\log_{10}$ of the $P$-value obtained using an inverse-variance meta- analysis) from the single-variant GWAS originally reported by Michailidou et al.[2]. **b** Single-variant GWAS adjusted for 212 sentinel risk variants and LD-score intercept; $P$-values were obtained with the GCTA-COJO joint analysis. **c** Gene-based analysis of adjusted GWAS results; $P$-values were obtained with the EUGENE gene-based test of association

**Estrogen receptor (ER)-negative breast cancer.** We applied the same analyses described above to results from the Milne et al. GWAS of ER-negative breast cancer, which included data on 21,468 cases and 100,594 controls, combined with 18,908 BRCA1 mutation carriers (9414 with breast cancer)[3].

Of the 54 sentinel risk variants identified through approximate joint association analysis (Supplementary Data 14), 19 were in LD ($r^2 > 0.8$) with a sentinel eQTL (Supplementary Data 15), implicating 24 genes as likely targets of risk-associated variants for ER-negative breast cancer (Supplementary Data 16). Of these, 13 were also identified as likely targets of variants associated with overall breast cancer risk, while the remaining 11 genes were specific to ER-negative risk variants: *ATM, CCNE1, CUL5, MCHR1, MDM4, NPAT, OCEL1, PIK3C2B, RALB, RP5-855D21.3,* and *WDR43*.

Seventeen genes were not highlighted as candidate target genes in the Milne et al. GWAS[3] (Supplementary Data 16 and Supplementary Data 17), mostly (15 genes) because they are

predicted targets of risk variants identified in previous GWAS, which were not considered by Milne et al.[3]. The two exceptions were *RP5-855D21.3* and *CUL5*, identified in our study based on eQTL from adipose tissue and whole-blood, respectively. Regional association plots for the 17 genes that represent previously unreported predictions are presented in Supplementary Fig. 3, with three examples shown in Fig. 4.

The disease protective allele was associated with lower gene expression for seven genes and higher gene expression for 11 genes (summary in Table 4 and Supplementary Data 18; detailed information in Supplementary Data 15); for the remaining six genes, directional effect was either not available (*ATM, CASP8, OCEL1, PEX14* and *WDR43*) or inconsistent across tissues (*ADCY3*).

Of the 24 target gene predictions, 18 were supported by the presence of enhancer– promoter chromatin interactions or an association between enhancer epigenetic marks and gene expression (Supplementary Data 19).

**Table 2 Risk loci for breast cancer identified in the EUGENE gene-based analysis but not in previous GWAS**

| Locus index | Gene | Chr | Start | N sentinel eQTL | | Gene-based P-value[a] | Sentinel eQTL with strongest association in the adjusted GWAS | | OncoScore |
|---|---|---|---|---|---|---|---|---|---|
| | | | | Tested | with P < 0.05 in adjusted GWAS[b] | | Variant | P-value[b] | |
| 1 | GSTM2 | 1 | 110210644 | 14 | 5 | 6.63E−08 | rs621414 | 4.08E−05 | 38.97 |
| 2 | SEMA4A | 1 | 156117157 | 9 | 5 | 1.45E−07 | rs887953 | 2.39E−06 | 27.04 |
| 3 | LINC00886 | 3 | 156465135 | 1 | 1 | 2.34E−06 | rs7641929 | 2.34E−06 | N/A |
| 4 | AC034220.3 | 5 | 131646978 | 7 | 4 | 5.92E−07 | rs11739622 | 0.000314 | N/A |
| 4 | IRF1 | 5 | 131817301 | 2 | 2 | 4.99E−07 | rs2548998 | 3.44E−05 | 42.46 |
| 5 | SIK2 | 11 | 111473115 | 2 | 2 | 4.09E−07 | rs527078 | 3.32E−05 | 39.57 |
| 5 | PPP2R1B | 11 | 111597632 | 8 | 2 | 2.31E−06 | rs680096 | 2.91E−06 | 56.52 |
| 6 | MAN2C1 | 15 | 75648133 | 14 | 6 | 1.91E−06 | rs8028277 | 2.16E−06 | 26.66 |
| 6 | RP11-817O13.8 | 15 | 75660496 | 4 | 4 | 3.83E−08 | rs4545784 | 3.85E−06 | N/A |
| 6 | SIN3A | 15 | 75661720 | 4 | 4 | 3.83E−08 | rs4545784 | 3.85E−06 | 42.43 |
| 6 | IMP3 | 15 | 75931426 | 1 | 1 | 2.30E−06 | rs4886708 | 2.30E−06 | 80.41 |

[a]Gene-based association P-value obtained when the EUGENE gene-based test was applied to the adjusted GWAS results
[b]P-value in the Michailidou et al. [2]. GWAS, adjusted for (i) the association with the sentinel risk variants identified in this study using the COJO-COND test; and (ii) the LD-score intercept

When we applied EUGENE to the ER-negative GWAS results obtained after conditioning on the 54 sentinel risk variants, we identified four genes in four loci with a significant gene-based association (EUGENE $P < 2.5 \times 10^{-6}$; Table 5, Supplementary Data 20 and Supplementary Fig. 4). Of these, we found that lower disease risk was consistently associated with lower expression for two genes (VPS52, GTF2IRD2B) and higher expression for one gene (INHBB). For the fourth gene (TNFSF10), directional effect was inconsistent across sentinel eQTL (detail and summary in Supplementary Tables 21 and 22, respectively).

Other genes that could be prioritized for functional follow-up include four (of the 24) target genes of sentinel risk variants that had a nominally significant gene-based association in the adjusted GWAS results (EUGENE $P < 0.05$; Supplementary Data 23): RALB, CCDC170, NPAT, and CASP8.

**Known role of the identified genes in cancer biology**. We used OncoScore, a text-mining tool that ranks genes according to their association with cancer based on available biomedical literature[21], to assess the extent to which each of the breast cancer genes we identified were already known to have a role in cancer. Of the 112 genes we identified across the overall and ER-negative analyses that could be scored by OncoScore, 48 scored below the recommended OncoScore cut-off threshold (21.09) for novelty, including 25 with an OncoScore of 0, indicating no prior evidence for a role in cancer biology (Tables 2 and 5; Supplementary Tables 3 and 16). For the remaining 64 genes there is an extensive literature on their role in cancer, and breast cancer in particular.

**Discussion**
To predict candidate target genes at breast cancer risk loci, we identified sentinel eQTL in multiple tissues that were in high LD ($r^2 > 0.8$) with sentinel risk variants from our recent GWAS[2]. Using this approach, we implicated 88 genes as likely targets of the overall breast cancer risk variants. Because eQTL are widespread, it is possible that some target gene predictions are false-positives due to coincidental overlap between sentinel eQTL and sentinel risk SNPs. At the LD threshold used, statistical methods developed recently to formally test for co-localization between eQTL and risk SNPs are of limited use, due to a high false-positive rate[22]. The 88 genes identified therefore represent target predictions that must be validated by functional studies. Of these 88, 26 genes had not been predicted as targets using a different approach that considered breast-specific functional annotations

and eQTL data[2], and so were considered previously unreported candidate target genes.

Of the 26 previously unreported target predictions, all but one were identified from eQTL analyses in blood, spleen, or immune cells. They include several genes with a known role in immunity, including: HLF, the expression of which is associated with the extent of lymphocytic infiltration after neo-adjuvant chemotherapy[23]; PTPN22, a shared autoimmunity gene[24], which encodes a protein tyrosine phosphatase that negatively regulates presentation of immune complex-derived antigens[25]; and RHBDD3, a negative regulator of TLR3-triggered natural killer cell activation[26], and critical regulator of dendritic cell activation[27]. In addition, we identified IRF1, which encodes a tumor suppressor and transcriptional regulator serving as an activator of genes involved in both innate and acquired immune response[28,29], as a previously unreported breast cancer risk locus. These results suggest that at least some of the previously unreported predicted target genes play a role in cancer cell elimination or inflammation. However, another possibility is that eQTL detected in well-powered studies of blood are predictors of eQTL in other less accessible tissues, including breast and adipose tissue. Consistent with this possibility, about 50% of the eQTL found to be in LD with a sentinel risk variant for overall breast cancer (and similarly for ER-negative breast cancer) were associated with the expression of the respective target gene in the relatively small GTEx breast tissue dataset, although not at the conservative threshold that we used to define sentinel eQTL. Of note, one previously unreported target was identified through eQTL analyses in adipose tissue: ZNF703. ZNF703 is a known oncogene in breast cancer[30], and has been reported to be associated with breast size[31] which might suggest a role in adiposity.

Using the same approach, we also identified 24 genes as likely targets of 19 ER-negative risk variants, of which 17 were not proposed as candidate target genes in the original GWAS[3]. Eleven of these 22 genes were unique to ER-negative breast cancer, including for example CUL5, a core component of multiple SCF-like ECS (Elongin-Cullin 2/5-SOCS-box protein) E3 ubiquitin-protein ligase complexes which recognize proteins for degradation and subsequent Class I mediated antigen presentation[32].

We also identified previously unreported breast cancer risk loci using the recently described EUGENE gene-based association test[6,7], which was developed to aggregate evidence for association with a disease or trait across multiple eQTL. Unlike other similar gene-based methods (e.g. S-PrediXcan), EUGENE includes in a single test information from eQTL identified in multiple tissues;

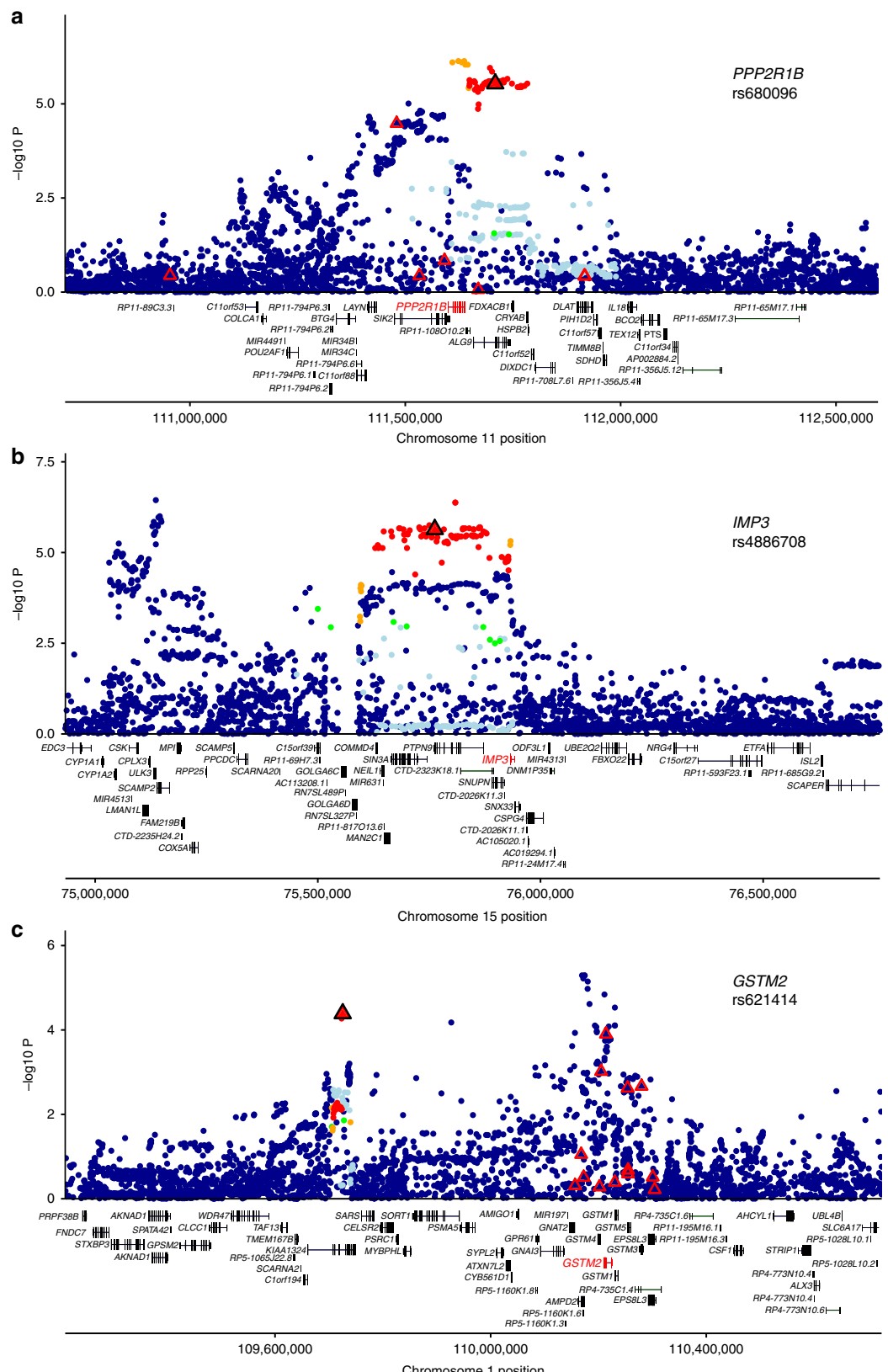

this property is expected to increase power to detect gene associations when multiple cell types/tissues contribute to disease pathophysiology, for two main reasons. First, because tissue-specific eQTL are common, and so a multi-tissue analysis is able to capture the association between all known eQTL and disease risk in a single test. Second, because in single-tissue analyses,

one needs to appropriately account for testing multiple tissues, thereby decreasing the significance threshold required for experiment-wide significance, which decreases power. When we applied EUGENE to the overall breast cancer GWAS[2], we identified 11 associated genes located in six previously unreported risk loci. For most of these genes, there were multiple sentinel

**Fig. 3** Examples of significant gene-based associations at loci not previously reported in breast cancer GWAS. Variants are represented by points colored according to the LD with the sentinel risk variant (red: ≥0.8, orange: 0.6–0.8, green: 0.4–0.6, light blue: 0.2–0.4, and dark blue: <0.2). Sentinel eQTL included in the EUGENE analysis (triangles) were identified from published eQTL studies of five different tissue types. Figure shows on the y-axis the evidence for breast cancer association (−log₁₀ of the P-value in the published GWAS after adjusting for the association with the sentinel risk variants using the COJO-COND test, and the LD-score intercept), and on the x-axis chromosomal position. The sentinel eQTL most associated with breast cancer risk is depicted by a black triangle; other sentinel eQTL included in the gene-based test are depicted by red triangles. Gene structures from GENCODE v19 gene annotations are shown and the predicted target genes shown in red. **a–c** show examples of three previously unreported loci which respectively implicate *PPP2R1B*, *IMP3* and *GSTM2* as candidate breast cancer susceptibility genes. Regional association plots for the remaining eight gene- based associations are provided in Supplementary Figure 2

**Table 3 Directional effect of genetically determined gene expression on disease risk for genes identified in the gene-based analysis of the adjusted breast cancer GWAS**

| Direction of effect | Predicted target genes of breast cancer sentinel risk variants |
|---|---|
| Decreased expression associated with decreased risk | *IMP3, IRF1, SEMA4A* |
| Increased expression associated with decreased risk | *LINC00886, MAN2C1, RP11-817O13.8, SIK2, SIN3A* |
| Ambiguous | *AC034220.3, GSTM2, PPP2R1B* |

eQTL associated with overall breast cancer risk. In the analysis of ER-negative breast cancer[3], EUGENE identified four associated genes (*INHBB, TNFSF10, VPS52,* and *GTF2IRD2B*) located in four previously unreported risk loci.

Some of the predicted target genes identified are well known to play a role in breast cancer carcinogenesis. For example, the genes identified for ER-negative breast cancer included *MDM4*, encoding a negative regulator of TP53, which is necessary for normal breast development[33]; *CCNE1*, an important oncogene in breast cancer[34,35]; *CASP8*, encoding a regulator of apoptosis[36]; *ATM*, a known breast cancer susceptibility gene[37,38]; and the ER, *ESR1*, which encodes a critical transcription factor in breast tissue[39]. On the other hand, the 11 significant gene-based associations for overall breast cancer included *GSTM2*, which is part of the mu class of glutathione S-transferases that are involved in increased susceptibility to environmental toxins and carcinogens[40]. Other noteworthy gene-based associations included those with: *IMP3*, which contributes to self-renewal and tumor initiation, properties associated with cancer stem cells[41]; *PPP2R1B*, which encodes the beta isoform of subunit A of Protein Phosphatase 2A, itself a tumor suppressor involved in modulating estrogen and androgen signaling in breast cancer[42]; and *SEMA4A*[43], recently shown to regulate the migration of cancer cells as well as dendritic cells[44].

Two recent studies reported results from analyses that are similar in scope to those carried out in our study. First, Hoffman et al.[45] reported that genetically determined expression of six genes was associated with risk of breast cancer: three when considering expression in breast tissue (*RCCD1, DHODH,* and *ANKLE1*) and three in whole blood (*RCCD1, ACAP1,* and *LRRC25*). Of note, *RCCD1* and *LRRC25* were identified as likely targets of known breast cancer risk variants in our analysis. We also found some support for an association between breast cancer risk and eQTL for *ACAP1* (EUGENE $P = 0.003$) and *ANKLE1* (EUGENE $P = 0.01$), but not for *DHODH* (best sentinel eQTL $P = 0.119$). Second, we recently applied a different gene-based approach called S-PrediXcan to results from the overall breast cancer GWAS[2], using gene expression levels predicted from breast tissue[20].

This study reported significant associations with 46 genes ($P < 5.82 \times 10^{-6}$), including 13 located in 10 regions not yet implicated by GWAS. A major difference between our analyses is that the latter were based on the original GWAS summary statistics, without adjusting for the effects of the sentinel risk

variants. This explains why most associated genes in their main analysis were located near known breast cancer risk variants. Of the 13 genes located in previously unreported risk loci, eight were tested in our analysis (which considered eQTL identified in multiple tissues, not just from breast as in ref. [20]), of which four had a nominally significant ($P < 0.05$) gene-based association: *MAN2C1* ($P = 1.9 \times 10^{-6}$), *SPATA18* ($P = 0.004$), *B3GNT1* ($P = 0.012$), and *CTD-2323K18.1* ($P = 0.021$). These results show that at least four of the associations reported by Wu et al.[20], which were based on information from breast eQTL only, are reproducible when a different gene-based approach is applied to the same GWAS results. Conversely, we identified a significant association with 13 genes not reported by Wu et al.[20], all with a gene-based association driven by eQTL identified in non-breast tissues, mostly in immune cells and/or whole-blood. For 78 of the 114 genes that we implicate in breast cancer risk, either through target gene prediction or gene-based analyses, we were able to determine the directional effect of the breast cancer protective alleles on gene expression. In some cases, this was consistent with their known function. For example, *ZNF703* is a well-known oncogene in breast cancer[30] and decreased expression was associated with decreased risk. Similarly, oncogenic activity has been reported for *PIK2C2B*[46], for which we found that decreased expression is associated with decreased risk. Another gene for which decreased expression was associated with decreased risk was *PTPN22* which is known to negatively regulate antigen presentation[47] and therefore might suppress immunoelimination. By contrast, *CCNE1*[48] and *APOBEC3A*[49] have been reported to have oncogenic roles, but we found that increased expression was associated with decreased risk. We have previously found the same counterintuitive relationship between breast cancer risk alleles and *CCND1* expression[50]. However, the expression patterns observed in breast tumors may not be relevant to the activity of these genes in the progenitor cells that give rise to breast tumors.

The directional effect of genetically determined gene expression on breast cancer risk is important because drugs that mimic the effect of the protective allele on gene expression might be expected to attenuate (rather than exacerbate) disease risk. For example, decreased risk of ER-negative breast cancer was associated with decreased expression of *KCNN4*, suggesting that an antagonist that targets this potassium channel and has a good safety profile[51] might reduce disease risk. Given these results, we suggest that *KCNN4* should be prioritized for functional and pre-clinical follow up.

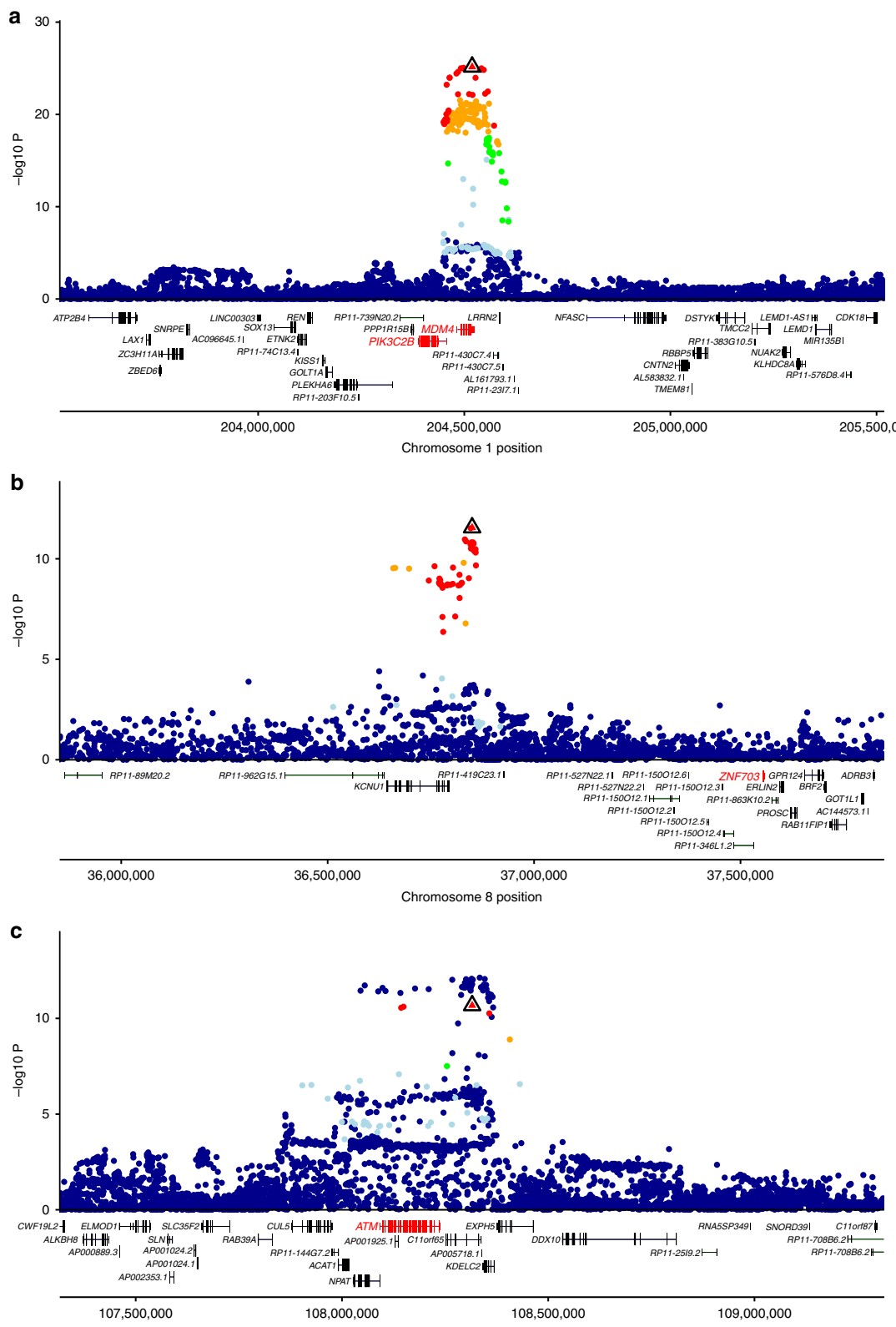

**Fig. 4** Examples of previously unreported target gene predictions at known ER- negative breast cancer risk loci. Variants are represented by points colored according to the LD with the sentinel risk variant (red: ≥0.8, orange: 0.6–0.8, green: 0.4–0.6, light blue: 0.2–0.4, and dark blue: <0.2). Sentinel risk variants (triangles) were identified based on joint association analysis[9]. Figure shows on the y-axis the evidence for ER-negative breast cancer association (−log$_{10}$ of the P-value in the original published GWAS results[3], obtained in that study using an inverse-variance meta-analysis), and on the x-axis chromosomal position. Gene structures from GENCODE v19 gene annotations are shown and the predicted target genes shown in red. The sentinel risk variants are in LD with sentinel eQTL for *MDM4* and *PIK3C2B* (**a**), *ZNF703* (**b**), and *ATM* (**c**; Supplementary Data 17). Regional association plots for the remaining 14 previously unreported target gene predictions are provided in Supplementary Figure 3

**Table 4 Directional effect of genetically-determined gene expression on disease risk for predicted target genes of ER-negative breast cancer sentinel risk variants**

| Direction of effect | Predicted target genes of breast cancer sentinel risk variants |
| --- | --- |
| Decreased expression associated with decreased risk | *CCDC170, DDA1, KCNN4, PIK3C2B, RP5- 855D21.3, SMG9, ZNF703* |
| Increased expression associated with decreased risk | *CCNE1, CENPO, CUL5, DNAJC27, ESR1, L3MBTL3, MCHR1, MDM4, NPAT, RALB, SYNE1* |
| Ambiguous | *ADCY3* |

**Table 5 Risk loci for ER-negative breast cancer identified in the EUGENE gene-based analysis and not in previous GWAS**

| Locus Index | Gene | Chr | Start | N sentinel eQTL | | Gene-based P-value[a] | Sentinel eQTL with strongest association in adjusted GWAS | | OncoScore |
| --- | --- | --- | --- | --- | --- | --- | --- | --- | --- |
| | | | | Tested | with P < 0.05 in adjusted GWAS* | | Variant | P-value[b] | |
| 1 | *INHBB* | 2 | 121103719 | 3 | 2 | 1.13E−07 | rs6542583 | 5.37E−06 | 25.82 |
| 2 | *TNFSF10* | 3 | 172223298 | 4 | 3 | 4.93E−07 | rs2041692 | 5.66E−07 | 86.01 |
| 3 | *VPS52* | 6 | 33218049 | 2 | 1 | 1.01E−06 | rs17215231 | 2.73E−07 | 17.45 |
| 4 | *GTF2IRD2B* | 7 | 74508364 | 1 | 1 | 8.52E−07 | rs2259337 | 8.52E−07 | 0 |

[a]Gene-based association P-value obtained when the EUGENE gene-based test was applied to the adjusted GWAS results
[b]P-value in the Milne et al. GWAS[3], adjusted for (i) the association with the sentinel risk variants identified in this study using the COJO-COND test; and (ii) the LD-score intercept

In summary, we have used the largest available GWAS of breast cancer, along with expression data from multiple different tissues, to identify 26 and 17 previously unreported likely target genes of known overall and ER-negative breast cancer risk variants, respectively. We also describe significant gene-based associations at six and four previously unreported risk loci for overall and ER-negative breast cancer, respectively. Further investigation into the function of the genes identified in breast and immune cells, particularly those which have additional support from experimental or computational predictions of chromatin looping, should provide additional insight into the etiology of breast cancer.

## Methods

**Predicting target genes of breast cancer risk variants**. Recently, Michailidou et al.[2] reported a breast cancer GWAS meta-analysis that combined results from 13 studies: the OncoArray study (61,282 cases and 45,494 controls); the iCOGS study (46,785 cases and 42,892 controls); and 11 other individual GWAS (with a combined 14,910 cases and 17,588 controls). That is, a total of 122,977 cases and 105,974 controls. The first aim of our study was to identify likely target genes of breast cancer risk variants identified in that GWAS.

First, we identified variants associated with variation in gene expression (i.e. eQTL) in published transcriptome studies of five broad tissue types: adipose, breast, immune cells isolated from peripheral blood, spleen and whole- blood. We identified a total of 35 transcriptome studies reporting results from eQTL analyses in any one of those five tissue types (Supplementary Data 2). Some studies included multiple cell types and/or experimental conditions, resulting in a total of 86 separate eQTL datasets. For each eQTL dataset, we then (i) downloaded the original publication tables containing results for the eQTL reported; (ii) extracted the variant identifier, gene name, association P-value and, if available, the effect size (specifically, by "effect size" we mean the beta/z-score) and corresponding allele; (iii) excluded eQTL located >1 Mb of the respective gene (i.e. *trans* eQTL), because often these are thought to be mediated by *cis* effects[52]; (iv) excluded eQTL with an association $P > 8.9 \times 10^{-10}$, a conservative threshold that corrects for 55,764 transcripts in Gencode v19, each tested for association with 1000 variants (as suggested by others[53–55]); and (v) for each gene, used the --clump procedure in PLINK to reduce the list of eQTL identified (which often included many correlated variants) to a set of 'sentinel eQTL', defined as the variants with strongest association with gene expression and in low LD ($r^2 < 0.05$, linkage disequilibrium (LD) window of 2 Mb) with each other.

Second, we identified variants that were independently associated with breast cancer risk at a $P < 5 \times 10^{-8}$ in the GWAS reported by Michailidou et al. [2], which included 122,977 cases and 105,974 controls. We refer to these as "sentinel risk variants" for breast cancer. To identify independent associations, we first excluded from the original GWAS (which tested 12,396,529 variants) variants with: (i) a sample size < 150,000; (ii) a minor allele frequency < 1%; (iii) not present in, or not polymorphic (Europeans) in, or with alleles that did not match, data from the 1000 Genomes project (release 20130502_v5a); and (iv) not present in, or with alleles

that did not match, data from the UK Biobank study[56]. After these exclusions, results were available for 8,248,946 variants. Next, we identified sentinel risk variants using the joint association analysis (--cojo- slct) option of GCTA[9], using imputed data from 5000 Europeans from the UK Biobank study[56] to calculate LD between variants. These individuals were selected based on the sample IDs (lowest 5000) from our approved UK Biobank application 25331.

Third, we identified genes for which a sentinel eQTL reported in any of the 86 eQTL datasets described above was in high LD ($r^2 > 0.8$) with a breast cancer sentinel risk variant. That is, we only considered genes for which there was high LD between a sentinel eQTL and a sentinel risk variant, which reduces the chance of spurious co-localization.

**Directional effect of gene expression on breast cancer risk**. Having identified a list of genes with expression levels correlated with sentinel risk variants, we then studied the directional effect of the breast cancer protective allele on gene expression. For each sentinel eQTL in high LD ($r^2 > 0.8$) with a sentinel risk variant, we: (i) identified the allele that was associated with reduced breast cancer risk, based on results reported by Michailidou et al. [2]; and (ii) determined if that allele was associated with increased or decreased target gene expression in each of the eQTL datasets that reported that eQTL. For many studies, the directional effect of eQTL (i.e. effect allele and beta) was not publicly available, and so for those this analysis could not be performed.

We also assessed whether the directional effect of gene expression on disease risk predicted by the approach described in the previous paragraph, which considered one eQTL at a time (a limitation) but many different eQTL datasets (a strength), would be recapitulated by applying S-PrediXcan[10] to the same breast cancer GWAS[2] using transcriptome information from 922 whole-blood samples studied by Battle et al.[11]. S-PrediXcan considers information from multiple eQTL identified for a given gene in a given tissue (e.g. whole-blood) when determining the association between genetically determined gene expression levels and disease risk. Therefore, we reasoned that this approach could be particularly useful for genes with multiple independent eQTL identified in the same tissue. The limitation of this approach is that it first requires the generation of gene expression prediction models based on individual-level variant and transcriptome data, which are not publicly available for most of the 35 transcriptome studies included in our analysis. We used gene expression models generated based on the whole-blood dataset of Battle et al.[11] because (i) most likely target genes were identified in our study based on eQTL information from whole-blood or immune cells isolated from whole-blood; and (ii) this was the largest transcriptome dataset we had access to.

**Target gene predictions supported by functional data**. Sentinels and variants in high LD ($r^2 > 0.8$ in Europeans of the 1000 Genomes Project, with MAF > 0.01) were queried against the following sources of publicly available data generated from blood-derived samples and cell lines. Computational methods linking regulatory elements with target genes including PreSTIGE[17], FANTOM5[16], IM-PET[18], enhancers and super enhancers from Hnisz et al.[19]. Experimental chromatin looping data defined by ChIA-PET[13] and capture Hi-C[4,14] and in situ Hi-C[15] were mined to identify physical interactions between query SNPs and target gene promoters. Variants were assigned to potential target genes based on intersection with associated enhancer annotations using BedTools intersect[57].

**Identification of previously unreported risk loci for breast cancer**. The second aim of this study was to use a gene-based approach to identify loci containing breast cancer risk variants that were missed by the single-variant analyses reported by Michailidou et al. [2]. At least three gene-based approaches have been described recently to combine in a single test the evidence for association with a disease across multiple eQTL[6,8,10]. Of these, we opted to use EUGENE[6,7] because it is applicable to GWAS summary statistics and combines in the same association test information from eQTL identified in different tissues and/or transcriptome studies. The latter feature is important for two main reasons. First, because multiple tissue types are likely to play a role in the pathophysiology of breast cancer, and tissue-specific eQTL are common[58]. Second, because different transcriptome studies of the same tissue (e.g. blood) identify partially (not completely) overlapping lists of eQTL. This might arise, for example, because of differences in sample size, gene expression quantification methods (e.g. microarrays vs. RNA-seq, data normalization) or demographics of the ascertained individuals (e.g. age, disease status). Therefore, identifying eQTL based on information from multiple tissues and/or studies is expected to produce a more comprehensive list of regulatory variants that could be relevant to breast cancer pathophysiology. An additional advantage of EUGENE is that it considers in the same test different types of eQTL (e.g. with exon-specific or stimulus-specific effects), thereby increasing the likelihood that causal regulatory variants related to breast cancer are captured in the analysis[59].

EUGENE requires an input file that lists all eQTL that will be included in the gene-based test for each gene. To generate such list for this study, we did as follows for each gene in the genome. First, we took the union of all eQTL reported in the 86 eQTL datasets described above. Second, we used the --clump procedure in PLINK to reduce the list of reported eQTL to a set of 'sentinel eQTL', defined as the variants with strongest association with gene expression and in low LD ($r^2 < 0.1$, LD window of 1 Mb) with each other. Note that clumping was not performed separately for each tissue or study, but rather applied to the union of eQTL identified across all tissues/studies. If an eQTL was identified in multiple tissues/studies, the clumping procedure was performed using the smallest $P$-value reported for that eQTL across all tissues/studies. A file (BREASTCANCER.20170517.eqtl.proxies.list) containing the sentinel eQTL identified per gene is available at [https://genepi.qimr.edu.au/staff/manuelF/eugene/main.html].

For each gene, EUGENE extracts single-variant association results for each sentinel eQTL identified (or, if not available, for a proxy with $r^2 > 0.8$) from the GWAS summary statistics, sums the association chi-square values across those eQTL, and estimates the significance (i.e. $P$-value) of the resulting sum test statistic using Satterthwaite's approximation, which accounts for the LD between eQTL[7]. This approximation was originally implemented by Bakshi et al.[60] in the GCTA-fastBAT module and is now also available in EUGENE. LD between eQTL was estimated based on data from 294 Europeans from the 1000 Genomes Project (release 20130502_v5a).

Because our aim was to identify previously unreported breast cancer risk loci, we did not apply EUGENE to the original results reported by Michailidou et al. [2]. Had we done so, significant gene-based associations would have been disproportionally located in known risk loci; associations driven by previously unreported risk variants would therefore be more difficult to highlight. Instead, we first adjusted the results[2] for the effects of the sentinel risk variants identified (see section above), using the --cojo-cond option of GCTA[9]. In doing so, we obtained adjusted GWAS results with no single variant with an association $P < 5 \times 10^{-8}$. We then applied EUGENE to the adjusted GWAS results, including a correction of the single-variant association statistic (i.e. chi-square) for an LD-score regression intercept[61] of 1.1072. This correction was important to account for the inflation of single-variant test statistics observed in Michailidou et al.[2] that were likely due to unaccounted biases.

To maintain the overall false-positive rate at 0.05, the significance threshold required to achieve experiment-wide significance in the gene-based analysis was set at $P < 0.05/N$ genes tested.

**OncoScore**. We used OncoScore, a text-mining tool that ranks genes according to their association with cancer based on available biomedical literature[21], to determine which of the breast cancer genes we identified were already known to have a role in cancer.

**ER-negative breast cancer**. Lastly, we used the approaches described above to identify target genes and previously unreported risk loci for ER-negative breast cancer. In this case, single-variant summary association statistics were obtained from the Milne et al.[3] GWAS, which included 21,468 ER-negative cases and 100,594 controls from the Breast Cancer Association Consortium, combined with 18,908 BRCA1 mutation carriers (9414 with breast cancer) from the Consortium of Investigators of Modifiers of BRCA1/2, all tested for 17,304,475 variants (9,827,195 after the exclusions described above). The LD-score regression intercept used to correct the single-variant association statistics of this GWAS was 1.0637.

**Study approval**. Informed consent was obtained from all subjects participating in the Breast Cancer Association Consortium under the approval of local Institutional Review Boards. Ethics approval was obtained from the Human Research Ethics Committee of QIMR-Berghofer.

## Data availability
GWAS summary statistics analyzed in this study are available upon request from the BCAC and CIMBA co-ordinators. The list of sentinel eQTL identified from publicly available datasets are available for download from https://genepi.qimr.edu.au/staff/manuelF/eugene/main.html.

## Code availability
The EUGENE gene-based approach is implemented in C++ and is available for download from https://genepi.qimr.edu.au/staff/manuelF/eugene/main.html.

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

## Acknowledgements

Full details of the Acknowledgements are provided in Supplementary Note 1.

## Author contributions

Writing Group: M.A.F., F. Al-E., W.Z., D.F.E., R.L.M., J.B., G.C.-T.; Statistical analysis: M.F. and E.R.G.; Provided DNA samples and/or phenotypic data: K.A., I.L.A., H.A.-C., A.C.A, A.A., V.A., K.J.A., B.K.A., E.A., J.A., J.B., D.R.B., D.B., M.W.B., S.B.,J.B., M.B., C.B., N.V.B., S.E.B., M.K.B., A.B., H.B., H.B., A.B., B.B., T.C., M.A.C., I.C., F.C., J.C., B.D.C., J.E.C., J.C.-C., D.C., H.C., W.K.C., K.B.M.C., C.L.C., F.J.C., A.C., S.J.C., S.S.C., K.C., M.B.D., M. de la H., J.D., P.D., O.D., T.D., A.M.D., M.D., D.M.E., B.E., C.E., C.E., M.E., P.A.F., O.F., H.F., E.F., D.F., M.G., M.G.-D., P.A.G., S.M.G., J.G., M.G.-C., J.A.G.-S., M.M.G., G.G.G., G.G., A.K.G., M.S.G., D.E.G., A.G.-N., M.H.G., J.G., P.G., C.A.H., P.H., U.H., W.H., J.H., F.B.L.H., A.H., R.N.H., J.L.H., P.J.H., K.H., E.N.I., C.I., M.J., A.J., P.J., R.J., R.C.J., E.M.J., N.J., M.E.J., V.J., K.B., B.Y.K., T.K., E.K., J.I.K., Y.-D.K., I.K., P.K., V.N.K., Y.L., D.L., C.L., G.L., J.L., F.L., S.L., J.L., J.T.L., J.L., E.M., A.M., S.M., D.M., L.M.,A.M., U.M., K.M., A.M., M.M., F.M., L.M., A.M.M., K.L.N., S.L.N., H.N., I.N., F.C.N., L.N.-Z., K.O., E.O., O.L.O., H.O., A.O., J.P., T.-W.P.-S., M.T.P., I.S.P., A.P.,P.P., P.D.P.P., D.P.K., B.P., N.P., P.R., J.R., G.R., H.A.R., E.S., K.S., E.J.S., M.K.S., R.K.S., P.S., X.-O.S., J.S., C.F.S., P.S., M.C.S., J.J.S., A.B.S., J.S., A.J.S., W.J.T., J.A.T., M.R.T., MB.T., A.Te., M.T., K.T., D.L.T., M.T., A.E.T., D.T., T.T., N.T., C.M.V., R.L.N., C.J.VA., A.M.W. van den O., E.J.VR., A.V., A.V., Q.W., B.W., J.N.W., C.W., R.W., X.R.Y., D.Y., A.Z., M.M., GEMO Study Collaborators, GC-HBOC study Collaborators, EMBRACE Collaborators, HEBON Investigators, BCFR Investigators and ABCTB Investigators.

## Additional information

**Competing interests:** The authors declare no competing interests.

Manuel A. Ferreira[1], Eric R. Gamazon[2,3], Fares Al-Ejeh[1], Kristiina Aittomäki[4], Irene L. Andrulis[5,6], Hoda Anton-Culver[7], Adalgeir Arason[8,9], Volker Arndt[10], Kristan J. Aronson[11], Banu K. Arun[12], Ella Asseryanis[13], Jacopo Azzollini[14], Judith Balmaña[15,16], Daniel R. Barnes[17], Daniel Barrowdale[17], Matthias W. Beckmann[18], Sabine Behrens[19], Javier Benitez[20,21], Marina Bermisheva[22], Katarzyna Białkowska[23], Carl Blomqvist[24,25], Natalia V. Bogdanova[26,27,28], Stig E. Bojesen[29,30,31], Manjeet K. Bolla[17], Ake Borg[32], Hiltrud Brauch[33,34,35], Hermann Brenner[10,35,36], Annegien Broeks[37], Barbara Burwinkel[38,39], Trinidad Caldés[40], Maria A. Caligo[41], Daniele Campa[19,42], Ian Campbell[43,44], Federico Canzian[45], Jonathan Carter[46], Brian D. Carter[47], Jose E. Castelao[48], Jenny Chang-Claude[19,49], Stephen J. Chanock[50], Hans Christiansen[26], Wendy K. Chung[51], Kathleen B.M. Claes[52], Christine L. Clarke[53], GC-HBOC Study Collaborators, GEMO Study Collaborators, EMBRACE Collaborators, Fergus J. Couch[54], Angela Cox[55], Simon S. Cross[56], Kamila Czene[57], Mary B. Daly[58], Miguel de la Hoya[40], Joe Dennis[17], Peter Devilee[59,60], Orland Diez[15,61], Thilo Dörk[27], Alison M. Dunning[62], Miriam Dwek[63], Diana M. Eccles[64], Bent Ejlertsen[65], Carolina Ellberg[66], Christoph Engel[67], Mikael Eriksson[57], Peter A. Fasching[18,68], Olivia Fletcher[69], Henrik Flyger[70], Eitan Friedman[71,72], Debra Frost[17], Marike Gabrielson[57], Manuela Gago-Dominguez[73,74], Patricia A. Ganz[75], Susan M. Gapstur[47], Judy Garber[76], Montserrat García-Closas[50,77], José A. García-Sáenz[78], Mia M. Gaudet[47], Graham G. Giles[79,80,81], Gord Glendon[5], Andrew K. Godwin[82], Mark S. Goldberg[83,84], David E. Goldgar[85], Anna González-Neira[21], Mark H. Greene[86], Jacek Gronwald[23], Pascal Guénel[87], Christopher A. Haiman[88], Per Hall[57,89], Ute Hamann[90], Wei He[57], Jane Heyworth[91], Frans B.L. Hogervorst[92], Antoinette Hollestelle[93], Robert N. Hoover[50], John L. Hopper[80], Peter J. Hulick[94,95], Keith Humphreys[57], Evgeny N. Imyanitov[96], HEBON Investigators, BCFR Investigators, ABCTB Investigators, Claudine Isaacs[97], Milena Jakimovska[98], Anna Jakubowska[23,99], Paul A. James[44,100], Ramunas Janavicius[101], Rachel C. Jankowitz[102], Esther M. John[103], Nichola Johnson[69], Vijai Joseph[104], Beth Y. Karlan[105], Elza Khusnutdinova[22,106], Johanna I. Kiiski[107], Yon-Dschun Ko[108], Michael E. Jones[109], Irene Konstantopoulou[110], Vessela N. Kristensen[111,112], Yael Laitman[71], Diether Lambrechts[113,114], Conxi Lazaro[115], Goska Leslie[17], Jenny Lester[105], Fabienne Lesueur[116,117,118], Sara Lindström[119,120], Jirong Long[121], Jennifer T. Loud[86], Jan Lubiński[23], Enes Makalic[80], Arto Mannermaa[122,123,124], Mehdi Manoochehri[90], Sara Margolin[89,125], Tabea Maurer[49], Dimitrios Mavroudis[126], Lesley McGuffog[17], Alfons Meindl[127], Usha Menon[128], Kyriaki Michailidou[17,129], Austin Miller[130], Marco Montagna[131], Fernando Moreno[78], Lidia Moserle[131], Anna Marie Mulligan[132,133], Katherine L. Nathanson[134], Susan L. Neuhausen[135], Heli Nevanlinna[107], Ines Nevelsteen[136], Finn C. Nielsen[137], Liene Nikitina-Zake[138], Robert L. Nussbaum[139], Kenneth Offit[104,140], Edith Olah[141], Olufunmilayo I. Olopade[142], Håkan Olsson[66], Ana Osorio[20,21], Janos Papp[141], Tjoung-Won Park-Simon[27], Michael T. Parsons[1], Inge Sokilde Pedersen[143,144,145], Ana Peixoto[146], Paolo Peterlongo[147], Paul D.P. Pharoah[17,62], Dijana Plaseska-Karanfilska[98], Bruce Poppe[52], Nadege Presneau[63], Paolo Radice[148], Johanna Rantala[149], Gad Rennert[150], Harvey A. Risch[151], Emmanouil Saloustros[152], Kristin Sanden[153], Elinor J. Sawyer[154], Marjanka K. Schmidt[37,155], Rita K. Schmutzler[156,157], Priyanka Sharma[158], Xiao-Ou Shu[121], Jacques Simard[159], Christian F. Singer[13], Penny Soucy[159], Melissa C. Southey[160,161], John J. Spinelli[162,163], Amanda B. Spurdle[1], Jennifer Stone[80,164], Anthony J. Swerdlow[109,165], William J. Tapper[64], Jack A. Taylor[166,167], Manuel R. Teixeira[146,168], Mary Beth Terry[169], Alex Teulé[170], Mads Thomassen[171], Kathrin Thöne[49], Darcy L. Thull[172], Marc Tischkowitz[173,174], Amanda E. Toland[175], Diana Torres[90,176], Thérèse Truong[87], Nadine Tung[177], Celine M. Vachon[178], Christi J. van Asperen[179], Ans M.W. van den Ouweland[180],

Elizabeth J. van Rensburg[181], Ana Vega[182], Alessandra Viel[183], Qin Wang[17], Barbara Wappenschmidt[156,157], Jeffrey N. Weitzel[184], Camilla Wendt[125], Robert Winqvist[185,186], Xiaohong R. Yang[50], Drakoulis Yannoukakos[110], Argyrios Ziogas[7], Peter Kraft[187,188], Antonis C. Antoniou[17], Wei Zheng[121], Douglas F. Easton[17,62], Roger L. Milne[79,80,160], Jonathan Beesley[1] & Georgia Chenevix-Trench[1]

[1]Department of Genetics and Computational Biology, QIMR Berghofer Medical Research Institute, Brisbane, QLD 4006, Australia. [2]Division of Genetic Medicine, Department of Medicine, Vanderbilt University, Nashville, TN 37235, USA. [3]Clare Hall, University of Cambridge, Cambridge CB3 9AL, UK. [4]Department of Clinical Genetics, Helsinki University Hospital, University of Helsinki, 00290 Helsinki, Finland. [5]Fred A. Litwin Center for Cancer Genetics, Lunenfeld-Tanenbaum Research Institute of Mount Sinai Hospital, Toronto, ON M5G 1X5, Canada. [6]Department of Molecular Genetics, University of Toronto, Toronto, ON M5S 1A8, Canada. [7]Department of Epidemiology, Genetic Epidemiology Research Institute, University of California Irvine, Irvine, CA, USA 92617. [8]Department of Pathology, Landspitali University Hospital, 101 Reykjavik, Iceland. [9]BMC (Biomedical Centre), Faculty of Medicine, University of Iceland, 101 Reykjavik, Iceland. [10]Division of Clinical Epidemiology and Aging Research, C070, German Cancer Research Center (DKFZ), 69120 Heidelberg, Germany. [11]Department of Public Health Sciences, and Cancer Research Institute, Queen's University, Kingston, ON K7L 3N6, Canada. [12]Department of Breast Medical Oncology, University of Texas MD Anderson Cancer Center, Houston, TX 77030, USA. [13]Dept of OB/GYN and Comprehensive Cancer Center, Medical University of Vienna, 1090 Vienna, Austria. [14]Unit of Medical Genetics, Department of Medical Oncology and Hematology, Fondazione IRCCS Istituto Nazionale dei Tumori (INT), 20133 Milan, Italy. [15]Oncogenetics Group, Vall dHebron Institute of Oncology (VHIO), 8035 Barcelona, Spain. [16]Department of Medical Oncology, Vall d'Hebron Institute of Oncology (VHIO), University Hospital, Vall d'Hebron, 08035 Barcelona, Spain. [17]Centre for Cancer Genetic Epidemiology, Department of Public Health and Primary Care, University of Cambridge, Cambridge CB1 8RN, UK. [18]Department of Gynecology and Obstetrics, Comprehensive Cancer Center ER-EMN, University Hospital Erlangen, Friedrich-Alexander-University Erlangen-Nuremberg, 91054 Erlangen, Germany. [19]Division of Cancer Epidemiology, German Cancer Research Center (DKFZ), 69120 Heidelberg, Germany. [20]Centro de Investigación en Red de Enfermedades Raras (CIBERER), 46010 Valencia, Spain. [21]Human Cancer Genetics Programme, Spanish National Cancer Research Centre (CNIO), 28029 Madrid, Spain. [22]Institute of Biochemistry and Genetics, Ufa Federal Research Centre of Russian Academy of Sciences, 450054 Ufa, Russia. [23]Department of Genetics and Pathology, Pomeranian Medical University, 71-252 Szczecin, Poland. [24]Department of Oncology, Helsinki University Hospital, University of Helsinki, Helsinki 00290, Finland. [25]Department of Oncology, Örebro University Hospital, 70185 Örebro, Sweden. [26]Department of Radiation Oncology, Hannover Medical School, 30625 Hannover, Germany. [27]Gynaecology Research Unit, Hannover Medical School, 30625 Hannover, Germany. [28]N.N. Alexandrov Research Institute of Oncology and Medical Radiology, 223040 Minsk, Belarus. [29]Copenhagen General Population Study, Herlev and Gentofte Hospital, Copenhagen University Hospital, 2730 Herlev, Denmark. [30]Department of Clinical Biochemistry, Herlev and Gentofte Hospital, Copenhagen University Hospital, 2730 Herlev, Denmark. [31]Faculty of Health and Medical Sciences, University of Copenhagen, 2200 Copenhagen, Denmark. [32]Department of Oncology, Lund University and Skåne University Hospital, 222 41 Lund, Sweden. [33]Dr. Margarete Fischer-Bosch-Institute of Clinical Pharmacology, 70376 Stuttgart, Germany. [34]University of Tübingen, 72074 Tübingen, Germany. [35]German Cancer Consortium (DKTK), German Cancer Research Center (DKFZ), 69120 Heidelberg, Germany. [36]Division of Preventive Oncology, German Cancer Research Center (DKFZ) and National Center for Tumor Diseases (NCT), 69120 Heidelberg, Germany. [37]Division of Molecular Pathology, The Netherlands Cancer Institute - Antoni van Leeuwenhoek Hospital, 1066 CX Amsterdam, The Netherlands. [38]Molecular Epidemiology Group, C080, German Cancer Research Center (DKFZ), 69120 Heidelberg, Germany. [39]Molecular Biology of Breast Cancer, University Womens Clinic Heidelberg, University of Heidelberg, 69120 Heidelberg, Germany. [40]Molecular Oncology Laboratory, CIBERONC, Hospital Clinico San Carlos, IdISSC (Instituto de Investigación Sanitaria del Hospital Clínico San Carlos), 28040 Madrid, Spain. [41]Section of Molecular Genetics, Dept. of Laboratory Medicine, University Hospital of Pisa, 56126 Pisa, Italy. [42]Department of Biology, University of Pisa, 56126 Pisa, Italy. [43]Research Department, Peter MacCallum Cancer Center, Melbourne, VIC 3000, Australia. [44]Sir Peter MacCallum Department of Oncology, The University of Melbourne, Melbourne, VIC 3000, Australia. [45]Genomic Epidemiology Group, German Cancer Research Center (DKFZ), 69120 Heidelberg, Germany. [46]Department of Gynaecological Oncology, Chris O'Brien Lifehouse and The University of Sydney, Camperdown, NSW 2050, Australia. [47]Behavioral and Epidemiology Research Group, American Cancer Society, Atlanta, GA, USA 30303. [48]Oncology and Genetics Unit, Instituto de Investigacion Sanitaria Galicia Sur (IISGS), Xerencia de Xestion Integrada de Vigo-SERGAS, 36312 Vigo, Spain. [49]Cancer Epidemiology Group, University Cancer Center Hamburg (UCCH), University Medical Center Hamburg-Eppendorf, 20246 Hamburg, Germany. [50]Division of Cancer Epidemiology and Genetics, National Cancer Institute, National Institutes of Health, Department of Health and Human Services, Bethesda, MD 20850, USA. [51]Departments of Pediatrics and Medicine, Columbia University, New York, NY 10032, USA. [52]Centre for Medical Genetics, Ghent University, Gent 9000, Belgium. [53]Westmead Institute for Medical Research, University of Sydney, Sydney, NSW 2145, Australia. [54]Department of Laboratory Medicine and Pathology, Mayo Clinic, Rochester, MN 55905, USA. [55]Sheffield Institute for Nucleic Acids (SInFoNiA), Department of Oncology and Metabolism, University of Sheffield, Sheffield S10 2TN, UK. [56]Academic Unit of Pathology, Department of Neuroscience, University of Sheffield, Sheffield S10 2TN, UK. [57]Department of Medical Epidemiology and Biostatistics, Karolinska Institutet, 171 65 Stockholm, Sweden. [58]Department of Clinical Genetics, Fox Chase Cancer Center, Philadelphia, PA 19111, USA. [59]Department of Pathology, Leiden University Medical Center, 2333 ZA Leiden, The Netherlands. [60]Department of Human Genetics, Leiden University Medical Center, 2333 ZA Leiden, The Netherlands. [61]Clinical and Molecular Genetics Area, University Hospital Vall dHebron, Barcelona 08035, Spain. [62]Centre for Cancer Genetic Epidemiology, Department of Oncology, University of Cambridge, Cambridge CB1 8RN, UK. [63]Department of Biomedical Sciences, Faculty of Science and Technology, University of Westminster, London W1B 2HW, UK. [64]Faculty of Medicine, University of Southampton, Southampton SO17 1BJ, UK. [65]Department of Oncology, Rigshospitalet, Copenhagen University Hospital, DK-2100 Copenhagen, Denmark. [66]Department of Cancer Epidemiology, Clinical Sciences, Lund University, 222 42 Lund, Sweden. [67]Institute for Medical Informatics, Statistics and Epidemiology, University of Leipzig, 04107 Leipzig, Germany. [68]David Geffen School of Medicine, Department of Medicine Division of Hematology and Oncology, University of California at Los Angeles, Los Angeles, CA 90095, USA. [69]The Breast Cancer Now Toby Robins Research Centre, The Institute of Cancer Research, London SW7 3RP, UK. [70]Department of Breast Surgery, Herlev and Gentofte Hospital, Copenhagen University Hospital, 2730 Herlev, Denmark. [71]The Susanne Levy Gertner Oncogenetics Unit, Chaim Sheba Medical Center, 52621 Ramat Gan, Israel. [72]Sackler Faculty of Medicine, Tel Aviv University, 69978 Ramat Aviv, Israel. [73]Genomic Medicine Group, Galician Foundation of Genomic Medicine, Instituto de Investigación Sanitaria de Santiago de Compostela (IDIS), Complejo Hospitalario Universitario de Santiago, SERGAS, 15706 Santiago de Compostela, Spain. [74]Moores Cancer Center, University of California San Diego, La Jolla, CA 92037, USA. [75]Schools of Medicine and Public Health, Division of Cancer Prevention & Control Research, Jonsson Comprehensive Cancer Centre, UCLA, Los Angeles, CA 90096-6900, USA. [76]Cancer Risk and Prevention Clinic, Dana-Farber Cancer Institute, Boston, MA 02215, USA. [77]Division of Genetics and Epidemiology,

Institute of Cancer Research, London SM2 5NG, UK. [78]Medical Oncology Department, Hospital Clínico San Carlos, Instituto de Investigación Sanitaria San Carlos (IdISSC), Centro Investigación Biomédica en Red de Cáncer (CIBERONC), 28040 Madrid, Spain. [79]Cancer Epidemiology & Intelligence Division, Cancer Council Victoria, Melbourne, VIC 3004, Australia. [80]Centre for Epidemiology and Biostatistics, Melbourne School of Population and Global Health, The University of Melbourne, Melbourne, VIC 3010, Australia. [81]Department of Epidemiology and Preventive Medicine, Monash University, Melbourne, VIC 3004, Australia. [82]Department of Pathology and Laboratory Medicine, Kansas University Medical Center, Kansas City, KS 66160, USA. [83]Department of Medicine, McGill University, Montréal, QC H4A 3J1, Canada. [84]Division of Clinical Epidemiology, Royal Victoria Hospital, McGill University, Montréal, QC H4A 3J1, Canada. [85]Department of Dermatology, Huntsman Cancer Institute, University of Utah School of Medicine, Salt Lake City, UT 84112, USA. [86]Clinical Genetics Branch, Division of Cancer Epidemiology and Genetics, National Cancer Institute, Bethesda, MD 20850-9772, USA. [87]Cancer & Environment Group, Center for Research in Epidemiology and Population Health (CESP), INSERM, University Paris-Sud, University Paris-Saclay, 94805 Villejuif, France. [88]Department of Preventive Medicine, Keck School of Medicine, University of Southern California, Los Angeles, CA 90033, USA. [89]Department of Oncology, Södersjukhuset, 118 83 Stockholm, Sweden. [90]Molecular Genetics of Breast Cancer, German Cancer Research Center (DKFZ), 69120 Heidelberg, Germany. [91]School of Population and Global Health, The University of Western Australia, Perth, WA 6009, Australia. [92]Family Cancer Clinic, The Netherlands Cancer Institute - Antoni van Leeuwenhoek hospital, Amsterdam 1066 CX, The Netherlands. [93]Department of Medical Oncology, Family Cancer Clinic, Erasmus MC Cancer Institute, Rotterdam 3015 CN, The Netherlands. [94]Center for Medical Genetics, NorthShore University HealthSystem, Evanston, IL 60201, USA. [95]The University of Chicago Pritzker School of Medicine, Chicago, IL 60637, USA. [96]N.N. Petrov Institute of Oncology. St., Petersburg 197758, Russia. [97]Lombardi Comprehensive Cancer Center, Georgetown University, Washington, DC 20007, USA. [98]Research Centre for Genetic Engineering and Biotechnology 'Georgi D. Efremov', Macedonian Academy of Sciences and Arts, Skopje 1000, Republic of Macedonia. [99]Independent Laboratory of Molecular Biology and Genetic Diagnostics, Pomeranian Medical University, Szczecin 71-252, Poland. [100]Parkville Familial Cancer Centre, Peter MacCallum Cancer Center, Melbourne, VIC 3000, Australia. [101]Hematology, oncology and transfusion medicine center, Dept. of Molecular and Regenerative Medicine, Vilnius University Hospital Santariskiu Clinics, Vilnius 08410, Lithuania. [102]Department of Medicine, Division of Hematology/Oncology, UPMC Hillman Cancer Center, University of Pittsburgh School of Medicine, Pittsburgh, PA 15232, USA. [103]Department of Medicine, Division of Oncology, Stanford Cancer Institute, Stanford University School of Medicine, Stanford, CA 94304, USA. [104]Clinical Genetics Research Lab, Department of Cancer Biology and Genetics, Memorial Sloan-Kettering Cancer Center, New York, NY 10065, USA. [105]David Geffen School of Medicine, Department of Obstetrics and Gynecology, University of California at Los Angeles, Los Angeles, CA 90095, USA. [106]Department of Genetics and Fundamental Medicine, Bashkir State Medical University, 450076 Ufa, Russia. [107]Department of Obstetrics and Gynecology, Helsinki University Hospital, University of Helsinki, Helsinki 00290, Finland. [108]Department of Internal Medicine, Evangelische Kliniken Bonn gGmbH, Johanniter Krankenhaus, Bonn 53177, Germany. [109]Division of Genetics and Epidemiology, The Institute of Cancer Research, London SM2 5NG, UK. [110]Molecular Diagnostics Laboratory, INRASTES, National Centre for Scientific Research 'Demokritos', Athens 15310, Greece. [111]Department of Cancer Genetics, Institute for Cancer Research, Oslo University Hospital-Radiumhospitalet, Oslo 0379, Norway. [112]Institute of Clinical Medicine, Faculty of Medicine, University of Oslo, Oslo 0450, Norway. [113]VIB Center for Cancer Biology, VIB, Leuven 3001, Belgium. [114]Laboratory for Translational Genetics, Department of Human Genetics, University of Leuven, Leuven 3000, Belgium. [115]Molecular Diagnostic Unit, Hereditary Cancer Program, ICO-IDIBELL (Bellvitge Biomedical Research Institute, Catalan Institute of Oncology), CIBERONC, Barcelona 08908, Spain. [116]Genetic Epidemiology of Cancer team, Inserm U900, Paris 75005, France. [117]Institut Curie, Paris 75005, France. [118]Mines ParisTech, Fontainebleau 77305, France. [119]Department of Epidemiology, University of Washington School of Public Health, Seattle, WA 98195, USA. [120]Public Health Sciences Division, Fred Hutchinson Cancer Research Center, Seattle, WA 98109, USA. [121]Division of Epidemiology, Department of Medicine, Vanderbilt Epidemiology Center, Vanderbilt-Ingram Cancer Center, Vanderbilt University School of Medicine, Nashville, TN 37232, USA. [122]Translational Cancer Research Area, University of Eastern Finland, Kuopio 70210, Finland. [123]Institute of Clinical Medicine, Pathology and Forensic Medicine, University of Eastern Finland, Kuopio 70210, Finland. [124]Imaging Center, Department of Clinical Pathology, Kuopio University Hospital, Kuopio 70210, Finland. [125]Department of Clinical Science and Education, Södersjukhuset, Karolinska Institutet, Stockholm 118 83, Sweden. [126]Department of Medical Oncology, University Hospital of Heraklion, Heraklion 711 10, Greece. [127]Department of Gynecology and Obstetrics, University of Munich, Campus Großhadern, Munich 81377, Germany. [128]MRC Clinical Trials Unit at UCL, Institute of Clinical Trials & Methodology, University College London, London WC1V 6LJ, UK. [129]Department of Electron Microscopy/Molecular Pathology and The Cyprus School of Molecular Medicine, The Cyprus Institute of Neurology & Genetics, Nicosia 1683, Cyprus. [130]NRG Oncology, Statistics and Data Management Center, Roswell Park Cancer Institute, Buffalo, NY 14263, USA. [131]Immunology and Molecular Oncology Unit, Veneto Institute of Oncology IOV - IRCCS, Padua 35128, Italy. [132]Department of Laboratory Medicine and Pathobiology, University of Toronto, Toronto, ON M5S 1A8, Canada. [133]Laboratory Medicine Program, University Health Network, Toronto, ON M5G 2C4, Canada. [134]Basser Center for BRCA, Abramson Cancer Center, University of Pennsylvania, Philadelphia, PA 19066, USA. [135]Department of Population Sciences, Beckman Research Institute of City of Hope, Duarte, CA 91010, USA. [136]Leuven Multidisciplinary Breast Center, Department of Oncology, Leuven Cancer Institute, University Hospitals Leuven, Leuven 3000, Belgium. [137]Center for Genomic Medicine, Rigshospitalet, Copenhagen University Hospital, Copenhagen DK-2100, Denmark. [138]Latvian Biomedical Research and Study Centre, Riga LV-1067, Latvia. [139]Cancer Genetics and Prevention Program, University of California San Francisco, San Francisco, CA 94143-1714, USA. [140]Clinical Genetics Service, Department of Medicine, Memorial Sloan-Kettering Cancer Center, New York, NY 10065, USA. [141]Department of Molecular Genetics, National Institute of Oncology, Budapest 1122, Hungary. [142]Center for Clinical Cancer Genetics, The University of Chicago, Chicago, IL 60637, USA. [143]Molecular Diagnostics, Aalborg University Hospital, Aalborg 9000, Denmark. [144]Clinical Cancer Research Center, Aalborg University Hospital, Aalborg 9000, Denmark. [145]Department of Clinical Medicine, Aalborg University, Aalborg 9000, Denmark. [146]Department of Genetics, Portuguese Oncology Institute, Porto 4220-072, Portugal. [147]Genome Diagnostics Program, IFOM - the FIRC (Italian Foundation for Cancer Research) Institute of Molecular Oncology, Milan 20139, Italy. [148]Unit of Molecular Bases of Genetic Risk and Genetic Testing, Department of Research, Fondazione IRCCS Istituto Nazionale dei Tumori (INT), Milan 20133, Italy. [149]Clinical Genetics, Karolinska Institutet, Stockholm 171 76, Sweden. [150]Clalit National Cancer Control Center, Carmel Medical Center and Technion Faculty of Medicine, Haifa 35254, Israel. [151]Chronic Disease Epidemiology, Yale School of Public Health, New Haven, CT 06510, USA. [152]Department of Oncology, University Hospital of Larissa, Larissa 411 10, Greece. [153]City of Hope Clinical Cancer Genetics Community Research Network, Duarte, CA 91010, USA. [154]Research Oncology, Guy's Hospital, King's College London, London SE1 9RT, UK. [155]Division of Psychosocial Research and Epidemiology, The Netherlands Cancer Institute - Antoni van Leeuwenhoek hospital, Amsterdam 1066 CX, The Netherlands. [156]Center for Hereditary Breast and Ovarian Cancer, Faculty of Medicine and University Hospital Cologne, University of Cologne, Cologne 50937, Germany. [157]Center for Integrated Oncology (CIO), Faculty of Medicine and University Hospital Cologne, University of Cologne, 50937 Cologne, Germany. [158]Department of Internal Medicine, Division of Oncology, University of Kansas Medical Center, Westwood, KS 66205, USA. [159]Genomics Center, Centre Hospitalier Universitaire de Québec – Université Laval, Research Center, Québec City, QC G1V 4G2, Canada. [160]Precision Medicine, School of Clinical Sciences at Monash Health, Monash University, Clayton, VIC 3168, Australia. [161]Department of Clinical Pathology, The University of Melbourne, Melbourne, VIC 3010, Australia. [162]Population Oncology, BC Cancer, Vancouver, BC V5Z 1G1, Canada. [163]School of Population and

Public Health, University of British Columbia, Vancouver, BC V6T 1Z4, Canada. [164]The Curtin UWA Centre for Genetic Origins of Health and Disease, Curtin University and University of Western Australia, Perth, WA 6000, Australia. [165]Division of Breast Cancer Research, The Institute of Cancer Research, London SW7 3RP, UK. [166]Epidemiology Branch, National Institute of Environmental Health Sciences, NIH, Research Triangle Park, NC 27709, USA. [167]Epigenetic and Stem Cell Biology Laboratory, National Institute of Environmental Health Sciences, NIH, Research Triangle Park, NC 27709, USA. [168]Biomedical Sciences Institute (ICBAS), University of Porto, Porto 4050-013, Portugal. [169]Department of Epidemiology, Mailman School of Public Health, Columbia University, New York, NY 10032, USA. [170]Genetic Counseling Unit, Hereditary Cancer Program, IDIBELL (Bellvitge Biomedical Research Institute),Catalan Institute of Oncology, CIBERONC, Barcelona 08908, Spain. [171]Department of Clinical Genetics, Odense University Hospital, Odence C 5000, Denmark. [172]Department of Medicine, Magee-Womens Hospital, University of Pittsburgh School of Medicine, Pittsburgh, PA 15213, USA. [173]Program in Cancer Genetics, Departments of Human Genetics and Oncology, McGill University, Montréal, QC H4A 3J1, Canada. [174]Department of Medical Genetics, University of Cambridge, Cambridge CB2 0QQ, UK. [175]Department of Cancer Biology and Genetics, The Ohio State University, Columbus, OH 43210, USA. [176]Institute of Human Genetics, Pontificia Universidad Javeriana, Bogota 110231, Colombia. [177]Department of Medical Oncology, Beth Israel Deaconess Medical Center, Boston, MA 02215, USA. [178]Department of Health Science Research, Division of Epidemiology, Mayo Clinic, Rochester, MN 55905, USA. [179]Department of Clinical Genetics, Leiden University Medical Center, Leiden 2333 ZA, The Netherlands. [180]Department of Clinical Genetics, Erasmus University Medical Center, Rotterdam 3015 CN, The Netherlands. [181]Department of Genetics, University of Pretoria, Arcadia 0007, South Africa. [182]Fundación Pública galega Medicina Xenómica-SERGAS, Grupo de Medicina Xenómica-USC, CIBERER, IDIS, Santiago de Compostela, Spain. [183]Division of Functional onco-genomics and genetics, Centro di Riferimento Oncologico di Aviano (CRO), IRCCS, Aviano 33081, Italy. [184]Clinical Cancer Genomics, City of Hope, Duarte, CA 91010, USA. [185]Laboratory of Cancer Genetics and Tumor Biology, Cancer and Translational Medicine Research Unit, Biocenter Oulu, University of Oulu, Oulu 90570, Finland. [186]Laboratory of Cancer Genetics and Tumor Biology, Northern Finland Laboratory Centre Oulu, Oulu 90570, Finland. [187]Program in Genetic Epidemiology and Statistical Genetics, Harvard T.H. Chan School of Public Health, Boston 02115 MA, USA. [188]Department of Epidemiology, Harvard T.H. Chan School of Public Health, Boston, MA 02115, USA

## GC-HBOC Study Collaborators

Norbert Arnold[221,222], Bernd Auber[223], Nadja Bogdanova-Markov[224], Julika Borde[156,157,225], Almuth Caliebe[226], Nina Ditsch[127], Bernd Dworniczak[224], Stefanie Engert[227], Ulrike Faust[228], Andrea Gehrig[229], Eric Hahnen[156,157], Jan Hauke[156,157], Julia Hentschel[230], Natalie Herold[156,157,225], Ellen Honisch[231], Walter Just[232], Karin Kast[233], Mirjam Larsen[156,157], Johannes Lemke[230], Huu Phuc Nguyen[228], Dieter Niederacher[231], Claus-Eric Ott[234], Konrad Platzer[230], Esther Pohl-Rescigno[156,157], Juliane Ramser[227], Kerstin Rhiem[156,157], Doris Steinemann[223], Christian Sutter[235], Raymonda Varon-Mateeva[234], Shan Wang-Gohrke[236] & Bernhard H.F. Weber[237]

[221]Department of Gynaecology and Obstetrics, University Hospital of Schleswig-Holstein, Campus Kiel, Christian-Albrechts University Kiel, Kiel 24118, Germany. [222]Institute of Clinical Molecular Biology, University Hospital of Schleswig-Holstein, Campus Kiel, Christian-Albrechts University Kiel, 24118 Kiel, Germany. [223]Institute of Human Genetics, Hannover Medical School, 30625 Hannover, Germany. [224]Institute of Human Genetics, University of Münster, Münster 48149, Germany. [225]Center for Molecular Medicine Cologne (CMMC), Faculty of Medicine and University Hospital Cologne, University of Cologne, Cologne 50931, Germany. [226]Institute of Human Genetics, University Hospital of Schleswig-Holstein, Campus Kiel, Christian-Albrechts University Kiel, Kiel 24118, Germany. [227]Division of Gynaecology and Obstetrics, Klinikum rechts der Isar der Technischen Universität München, Munich 80333, Germany. [228]Institute of Medical Genetics and Applied Genomics, University of Tübingen, Tübingen 72074, Germany. [229]Department of Human Genetics, University Würzburg, Würzburg 97074, Germany. [230]Institute of Human Genetics, University Hospital Leipzig, Leipzig 04103, Germany. [231]Department of Gynecology and Obstetrics, University Hospital Düsseldorf, Heinrich-Heine University Düsseldorf, Düsseldorf 40225, Germany. [232]Institute of Human Genetics, University Hospital Ulm, Ulm 89075, Germany. [233]Department of Gynecology and Obstetrics, Technical University of Dresden, Dresden 01307, Germany. [234]Institute of Human Genetics, Campus Virchov Klinikum, Charite, Berlin 13353, Germany. [235]Institute of Human Genetics, University Hospital Heidelberg, 69120 Heidelberg, Germany. [236]Department of Gynaecology and Obstetrics, University Hospital Ulm, 89075 Ulm, Germany. [237]Institute of Human Genetics, University Regensburg, 93053 Regensberg, Germany

## GEMO Study Collaborators

Fabienne Prieur[238], Pascal Pujol[239], Charlotte Sagne[240], Nicolas Sevenet[241], Hagay Sobol[242,243], Johanna Sokolowska[244], Dominique Stoppa-Lyonnet[245,246,247] & Laurence Venat-Bouvet[248]

[238]Service de Génétique Clinique Chromosomique et Moléculaire, Hôpital Nord, CHU Saint Etienne, 42270 St. Etienne, France. [239]Unité d'Oncogénétique, CHU Arnaud de Villeneuve, 34295 Montpellier, France. [240]Bâtiment Cheney D, Centre Léon Bérard, Lyon 69373, France. [241]Oncogénétique, Institut Bergonié, Bordeaux 33076, France. [242]Département Oncologie Génétique, Prévention et Dépistage, Institut Paoli-Calmettes, Marseille 13009, France. [243]Marseille Medical School, Aix-Marseille University, Marseille 13007, France. [244]Laboratoire de génétique médicale, Nancy Université, Centre Hospitalier Régional et Universitaire, Vandoeuvre-les-Nancy 54511, France. [245]Service de Génétique, Institut Curie, Paris 75005, France. [246]Department of Tumour Biology, INSERM U830, 75005 Paris, France. [247]Université Paris Descartes, 75006 Paris, France. [248]Department of Medical Oncology, CHU Dupuytren, Limoges 87042, France

## EMBRACE Collaborators

Julian Adlard[195], Munaza Ahmed[196], Julian Barwell[197], Angela Brady[198], Carole Brewer[199], Jackie Cook[200], Rosemarie Davidson[201], Alan Donaldson[202], Jacqueline Eason[203], Ros Eeles[204], D. Gareth Evans[205,206], Helen Gregory[207], Helen Hanson[208], Alex Henderson[209], Shirley Hodgson[210], Louise Izatt[211], M. John Kennedy[212], Fiona Lalloo[206], Clare Miller[213], Patrick J. Morrison[214], Kai-ren Ong[215], Jo Perkins[17], Mary E. Porteous[216], Mark T. Rogers[217], Lucy E. Side[218], Katie Snape[219], Lisa Walker[220] & Patricia A. Harrington[62]

[195]Yorkshire Regional Genetics Service, Chapel Allerton Hospital, Leeds LS7 4SA, UK. [196]North East Thames Regional Genetics Service, Great Ormond Street Hospital for Children NHS Trust, London WC1N 3JH, UK. [197]Leicestershire Clinical Genetics Service, University Hospitals of Leicester NHS Trust, Leicester LE1 5WW, UK. [198]North West Thames Regional Genetics Service, Kennedy Galton Centre, The North West London Hospitals NHS Trust, Middlesex HA1 3UJ, UK. [199]Department of Clinical Genetics, Royal Devon & Exeter Hospital, Exeter EX2 5DW, UK. [200]Sheffield Clinical Genetics Service, Sheffield Children's Hospital, Sheffield S10 2TH, UK. [201]Department of Clinical Genetics, South Glasgow University Hospitals, Glasgow G51 4TF, UK. [202]Clinical Genetics Department, St. Michael's Hospital, Bristol BS2 8EG, UK. [203]Nottingham Clinical Genetics Service, Nottingham University Hospitals NHS Trust, Nottingham NG5 1PB, UK. [204]Oncogenetics Team, The Institute of Cancer Research and Royal Marsden NHS Foundation Trust, London SM2 5NG, UK. [205]Genomic Medicine, Division of Evolution and Genomic Sciences, The University of Manchester, Manchester Academic Health Science Centre, Manchester Universities Foundation Trust, St. Mary's Hospital, Manchester M13 9WL, UK. [206]Genomic Medicine, North West Genomics hub, Manchester Academic Health Science Centre, Manchester Universities Foundation Trust, St. Mary's Hospital, Manchester M13 9WL, UK. [207]North of Scotland Regional Genetics Service, NHS Grampian & University of Aberdeen, Foresterhill, Aberdeen, UK. [208]Southwest Thames Regional Genetics Service, St. George's Hospital, London SW17 0QT, UK. [209]Institute of Genetic Medicine, Centre for Life, Newcastle Upon Tyne Hospitals NHS Trust, Newcastle upon Tyne NE1 3BZ, UK. [210]Department of Clinical Genetics, St George's, University of London, London SW17 0RE, UK. [211]Clinical Genetics, Guy's and St Thomas' NHS Foundation Trust, London SE1 9RT, UK. [212]Academic Unit of Clinical and Molecular Oncology, Trinity College Dublin and St. James's Hospital, Dublin, Eire 8, Ireland. [213]Department of Clinical Genetics, Alder Hey Hospital, Liverpool L12 2AP, UK. [214]Northern Ireland Regional Genetics Centre, Belfast City Hospital, Belfast BT9 7AB, UK. [215]West Midlands Regional Genetics Service, Birmingham Women's Hospital Healthcare NHS Trust, Birmingham B15 2TG, UK. [216]South East of Scotland Regional Genetics Service, Western General Hospital, Edinburgh EH4 2XU, UK. [217]All Wales Medical Genetics Services, University Hospital of Wales, Cardiff CF14 4XW, UK. [218]Princess Anne Hospital, Southampton SO16 5YA, UK. [219]Medical Genetics Unit, St. George's, University of London, London SW17 0RE, UK. [220]Oxford Regional Genetics Service, Churchill Hospital, Oxford OX3 7LJ, UK

## HEBON Investigators

Bernadette A.M. Heemskerk-Gerritsen[93], Matti A. Rookus[249], Caroline M. Seynaeve[93], Frederieke H. van der Baan[249], Annemieke H. van der Hout[250], Lizet E. van der Kolk[92], Rob B. van der Luijt[251], Carolien H.M. van Deurzen[252], Helena C. van Doorn[253], Klaartje van Engelen[254], Liselotte van Hest[255], Theo A.M. van Os[256], Senno Verhoef[92], Maartje J. Vogel[257] & Juul T. Wijnen[258]

[249]Department of Epidemiology, The Netherlands Cancer Institute, Amsterdam 1066 CX, The Netherlands. [250]Department of Genetics, University Medical Center Groningen, University Groningen, Groningen 9713 GZ, The Netherlands. [251]Department of Medical Genetics, University Medical Center, Utrecht 3594 CX, The Netherlands. [252]Department of Pathology, Erasmus University Medical Center, Rotterdam 3015 CN, The Netherlands. [253]Department of Gynaecology, Family Cancer Clinic, Erasmus MC Cancer Institute, Rotterdam 3015 CE, The Netherlands. [254]Department of Clinical Genetics, VU University Medical Center, Amsterdam 1105 AZ, The Netherlands. [255]Clinical Genetics, Amsterdam UMC, Vrije Universiteit Amsterdam, Amsterdam 1007 MB, The Netherlands. [256]Department of Clinical Genetics, Amsterdam UMC, location AMC, Amsterdam 1100 DD, The Netherlands. [257]Department of Pathology, Netherlands Cancer Institute, Amsterdam 1006 BE, The Netherlands. [258]Department of Human Genetics and Department of Clinical Genetics, Leiden University Medical Center, Leiden 2333 ZA, The Netherlands

## BCFR Investigators

Alexander Miron[189,190], Miroslav Kapuscinski[80], Anita Bane[191], Eric Ross[192], Saundra S. Buys[193] & Thomas A. Conner[194]

[189]Department of Cancer Biology, Dana-Farber Cancer Institute, Boston, MA 02215, USA. [190]Department of Surgery, Harvard Medical School, Boston, MA 02215, USA. [191]Department of Pathology & Molecular Medicine, Juravinski Hospital and Cancer Centre, McMaster University, Hamilton, ON L8V 1C3, Canada. [192]Population Studies Facility, Fox Chase Cancer Center, Philadelphia, PA 19111, USA. [193]Department of Medicine, Huntsman Cancer Institute, Salt Lake City, UT 84112, USA. [194]Huntsman Cancer Institute, Salt Lake City, UT 84112, USA

## ABCTB Investigators

Rosemary Balleine[259], Robert Baxter[260], Stephen Braye[261], Jane Carpenter[262], Jane Dahlstrom[263], John Forbes[264], Soon C Lee[265], Deborah Marsh[260], Adrienne Morey[266], Nirmala Pathmanathan[267], Peter Simpson[268], Allan Spigelman[269], Nicholas Wilcken[270] & Desmond Yip[271]

[259]Pathology West ICPMR, Westmead, NSW 2145, Australia. [260]Kolling Institute of Medical Research, University of Sydney, Royal North Shore Hospital, Sydney, NSW 2065, Australia. [261]Pathology North, John Hunter Hospital, Newcastle, NSW 2305, Australia. [262]Australian Breast Cancer Tissue Bank, Westmead Institute for Medical Research, University of Sydney, Sydney, NSW 2145, Australia. [263]Department of Anatomical Pathology, ACT Pathology, ACT Pathology, Canberra Hospital and ANU Medical School, Australian National University, Canberra, ACT 2605, Australia. [264]Department of Surgical Oncology, Calvary Mater Newcastle Hospital, Australian New Zealand Breast Cancer Trials Group, and School of Medicine and Public Health, University of Newcastle, Newcastle, NSW 2035, Australia. [265]School of Science and Health, The University of Western Sydney, Sydney, NSW 2650, Australia. [266]SydPath St. Vincent's Hospital, Sydney, NSW 2010, Australia. [267]Department of Tissue Pathology and Diagnostic Oncology, Pathology West, Westmead Breast Cancer Institute, Westmead Hospital, Sydney, NSW 2145, Australia. [268]UQ Centre for Clinical Research and School of Medicine, The University of Queensland, Brisbane, QLD 4072, Australia. [269]Hereditary Cancer Clinic, St. Vincent's Hospital, The Kinghorn Cancer Centre, Sydney, NSW 2010, Australia. [270]Crown Princess Mary Cancer Centre, Westmead Hospital, Sydney, NSW 2145, Australia. [271]Department of Medical Oncology, The Canberra Hospital, Canberra, ACT 2605, Australia

