## [Peer Review File · Nature Communications]

Reviewers' comments:

Reviewer #1 (Remarks to the Author):

The authors studied the association of GWAS-derived breast cancer susceptibility loci with gene expression patterns (eQTL-based) in a large cohort of 122,977 cases and 105,974 controls using a previously published analysis pipeline, termed INQUISIT. As the major novelty that separates this study from the previous report in Nature 2017, they added eQTL datasets from tissue types other than breast cancer: adipose, individual immune cell types, spleen, and whole-blood. Using this approach, they report – as the main finding - 18 novel candidate target genes whose expression is associated with overall breast cancer risk and 22 candidate target genes (15 novel) whose expression is associated with the risk of estrogen receptor-negative breast cancer, based on their relationship with a subset of 212 breast cancer risk variants (that were identified from a joint analysis in GCTA). They also used EUGENE to describe gene-based associations. It is the implication from this study that the described novel associations may reflect genetically-determined gene expression in tissues other than breast epithelium, like immune cells, and the gene expression in these tissues may define breast cancer risk. However, this is just speculation at this time.

The authors provide a rational design and analyzed a large number of samples to conduct the analysis. The results are promising but need further interpretation. A case in point functional work would be helpful to better support the predictions. Nevertheless, in its current form, the manuscript is somewhat confusing and not easily to follow, and misses tables 2-7 and all supplementary tables. It is also not easy to pair figure legends to the figures.

How exactly were the 212 risk variants identified? How do we know GCTA?

What eQTL datasets from tissues other than breast cancer were incorporated into the analysis? Are they sufficiently powered? From reading the manuscript, only the analysis of eQTL dataset related to whole blood is described sufficiently, and seem to have generated useful data. Are there significant associations of the 212 loci (SNPs) with gene expression in tissues other than breast that are not found in breast tissue – indicating that genetically-determined expression in immune cells or adipocyte is the driver of breast cancer risk? Having these associations in both breast and the other tissues may not be as informative to support a driver role of genetically-determined expression in immune cells or adipocyte.

Comments:

Author identified 212 variants associated with breast cancer using the GCTA GWAS (supplementary table 1, which the reviewer could not find). As I understand this GWAS composes of the "original GWAS" (14910 cases and 17588 controls), iCOGS, and the "OncoArray" (61282 cases and 45494 controls) datasets. Not clear. Do all 212 SNPs have the same direction of risk in all three studies? Do the 20 SNPs in the joint analysis contain flipped alleles?

For sentinel 46 eQTL SNPs, author set the 1Mb gene boundary and r^2 to less than 0.05, and use a conservative p value 2.3×10^{-9} , however, for EUGENE the setting was r^2 0.1 and window size of 2Mb. When they apply settings for EUGENE, should the authors not use the same parameters? Otherwise, the authors need to discuss the effect of those parameters changes on the final results.

Also for the sentinel eQTL SNPs, are those low frequency SNPs, what is the MAF distribution of them? Should a setting for effect size be applied? What would be the implication of low frequency SNPs?

Authors queried 86 eQTL datasets (supplementary table 2, could not see it), what is the sample size for the different tissues (adipose, breast, immune cells, spleen and whole blood (blood

monocytes?), are they very different, and the whole blood datasets are the major source? What is whole blood? PBMCs?

For the direction effect of gene expression on cancer risk, should author consider the known oncogenic or tumor suppressor feature of the genes? The allele effect on gene expression should be consistent with known oncogene or tumor suppressor function, or should be discussed if different.

Author applied S-PrediXscan using 922 whole blood samples from the Battle dataset. This dataset came from a study of depression with 463 subjects with depression and 459 subjects without it. Is there a concern of selection bias?

Why did the authors not apply EUGENE to each tissue? Wouldn't tissue-specific data be informative?

Gene-based tests and eQTL analysis led to similar findings, correct?

Page 17, Oncoscore analysis. The authors analyzed 47 genes. Why 47 genes?

Page 22, targeting KCNN4. What is the magnitude of association of the genetically-determined gene expression with breast cancer risk? Would it make sense to develop a drug that targets KCNN4 for disease prevention? All drugs have also side effects.

Reviewer #2 (Remarks to the Author):

The aim of this study is to identify candidate genes for increased risk of breast cancer from existing databases which identified high risk loci. This study specifically focused on breast and non-breast tissues.

Identifying specific genes which might contribute to risk of development of breast cancer is an important achievement for confirming known genes involved in breast cancer biology but also and maybe more importantly a hypothesis generating discovery of genes not previously described as associated with breast cancer or with unknown function. These results can enhance the basic science discovery and are important addition to the public domain.

In addition, the authors attempted to identify novel loci containing breast cancer risk variants that were missed in previous studies.

The statistical tools used in this publication seem to be described previously and used for similar approaches but require a review by biostatistician to make sure they have been used in the most appropriate manner. Authors would strengthen the publication by including a more robust discussion of why they believe the tools they chose are the most appropriate and thus reveal biologically accurate results.

I believe that the current article adds a significant contribution to the breast cancer research community and could assist with ongoing research programs and lead to new research avenues not previously pursued.

Reviewer #3 (Remarks to the Author):

The manuscript by Ferreira et al. represents an extensive effort that combines several large-scale

publicly-available data sets: 86 expression quantitative trait locus (eQTL) analysis data sets derived from blood, immune, spleen, adipose, and normal breast cells together with summary statistics from the largest GWAS of breast cancer susceptibility published so far. Several putative target genes of previously identified breast cancer risk loci are uncovered and some of these may be new candidate susceptibility genes for overall and/or ER-negative breast cancer. A transcriptome-wide association study (TWAS) using the EUGENE method (developed and previously published by the lead author) is also performed to identify new potential breast cancer risk loci that are distant from previously identified loci. Strengths of the analyses include the application of EUGENE, the study of multiple independent eQTLs for the same gene that are usually ignored by other researchers, and the adjustment of the original GWAS summary statistics for associations at the known risk loci, which ensures that the EUGENE results are not driven by the known risk loci (unlike other TWAS). While disease-specific risk variants may exert their functional effects by modulating gene expression in a tissue-specific manner it is not known whether they may have such modulatory effects in other tissue types or if a tissue type with more samples available for eQTL analysis, such as blood, may be leveraged for linking variants to gene expression. The study by Ferreira et al. is likely to be of substantial interest to the genetics community since linking GWAS-identified risk variants to genes is a challenging first step in the biomedical translation of GWAS.

Specific comments:

Main Table 1 presents, "Novel target gene predictions of sentinel risk variants for overall breast cancer." It appears that at known GWAS-identified risk loci for overall breast cancer, the same sentinel risk SNPs or SNPs in strong LD with these SNPs are often predicted to target more than one gene. For example, when the first row of main Table 1 is read in conjunction with Supplementary Table 4 and the corresponding regional association plot, at the 2p23.3 locus alone, there are four predicted targets – ADCY3, CENPO, DNAJC27, and RP11-443B20.1.

- How is this to be interpreted in the absence of experimental evidence?
- Are the effects of the sentinel risk SNPs on breast cancer mediated by the expression of all four genes – is such multiplicity of genetic regulation plausible? Is this simply an artefact due to nearby genes being co-expressed?
- Should one use the OncoScore to prioritise CENPO, which has the higher OncoScore, over the other three genes?
- Do all four eQTL signals colocalise with the sentinel risk SNPs or can colocalisation be used to further dissect this and other similar loci?

Further, there is strong evidence for an eQTL in TCGA breast cancer tissue for ADCY3 that colocalises with the sentinel/credible causal breast cancer risk SNPs at this locus (Supplementary Table 18 in Michailidou et al. Nature 2017; column P: risk esnp p/best esnp p = 28). Isn't a tissue-specific eQTL that colocalises more likely to represent the true target gene given that there is also evidence favouring ADCY3 from the authors' present analysis and another recent breast tissue-specific eQTL analysis of breast cancer risk loci (Guo et al. AJHG 2018)? Likewise, there is strong eQTL and colocalisation evidence for AMFR at the 16q12.2 locus in normal breast tissues from the authors' 2017 paper and the current analysis and it is quite possible that the novel gene highlighted by the authors at this locus (CES1) is a false positive given that it also has a lower OncoScore than AMFR. Along the same lines, it is very difficult to choose between STXBP4 and HLF with the breast-specific evidence pointing to the former. All the above limitations should be highlighted and discussed. Far more caution is urged in the presentation of these results as novel findings implicating immunological regulation at breast cancer risk loci rather than the result of routine co-expression of neighbouring genes in eukaryotic genomes being picked up by the use of dozens of eQTL data sets all together (see for example Michalak, Genomics 2008).

It may be helpful for the reader to weigh the novelty of the eQTL evidence for themselves if all loci in Table 1 where the novel target gene prediction is only one of multiple (please specific the

number) target genes predicted at the same locus are flagged in a separate column. A truly “novel” gene in the context of the authors’ aims (Introduction, pages 11 and 12) might be one where there is no reasonable breast tissue-specific eQTL prediction but where the authors’ multi-tissue eQTL prediction has revealed a target gene. Please highlight in Table 1 the target genes with breast tissue-specific eQTL evidence (if any) at such loci. The same comments also apply to Table 5.

Why was the EUGENE method not applied to the known breast cancer risk loci? It would be interesting to compare the EUGENE predictions for the known breast cancer risk loci to the authors’ eQTL-based predictions for the same loci. While I accept the authors’ rationale for applying EUGENE to adjusted GWAS summary statistics for the identification of completely new breast cancer risk loci (Methods, page 27), this does not preclude the application of EUGENE to the original GWAS summary statistics at the known loci for comparison against eQTL findings.

Since OncoScore uses text-mining of the published biomedical literature to link gene names with cancer, would it not be contaminated by the fact that several of these genes already find mention in breast cancer GWAS papers due to their proximity to the reported risk loci for this cancer? This could lead to spuriously elevated scores for some genes and its influence on the OncoScore is not clear to me. Would a look-up of gene essentiality in cancer or immune cell lines provide a more unbiased measure of carcinogenic/immune regulatory potential?

The question about multiple target genes in the same putative breast cancer susceptibility region also arises for the EUGENE results in Table 3, in particular for the three GSTMx transcripts at 1p13.3 that are simply likely to be co-expressed together. The ultimate focus of EUGENE on identifying sub-GWAS-significant risk loci rather than the current emphasis on the biological function of specific genes should perhaps be reiterated in the discussion.

Discussion, page 19: “However, another possibility is that eQTLs detected in well-powered studies of blood are predictors of eQTL in other less accessible tissues, including breast and adipose tissue.” Since the authors use 183 GTEx normal breast tissue samples in their analysis could this possibility be formally tested? It may be helpful to those broadly interested in the functional follow-up of GWAS to know how frequently do eQTLs identified in the largest blood eQTL data set analysed by the authors also show up at less significant p-value thresholds in the GTEx normal breast tissues.

The statistical analyses in this manuscript are appropriate and valid and given the level of methodological detail provided, I believe that the results of the manuscript can be reproduced.

Reviewer #1 (Remarks to the Author):

1. It is the implication from this study that the described novel associations may reflect genetically-determined gene expression in tissues other than breast epithelium, like immune cells, and the gene expression in these tissues may define breast cancer risk. However, this is just speculation at this time.

We have changed the discussion to include the word 'may' as follows: "These results suggest that at least some of the novel genes may play a role in cancer cell elimination or inflammation" and further modified this paragraph to respond to Reviewer 3 (see below).

2. The authors provide a rational design and analyzed a large number of samples to conduct the analysis. The results are promising but need further interpretation. A case in point functional work would be helpful to better support the predictions.

We have now highlighted genes (in Supplementary Tables 8 and 17) for which our target prediction based on eQTL data is supported by publicly available functional data, specifically (i) chromatin interactions between an enhancer and the gene promoter; or (ii) an association between variation in enhancer epigenetic marks and variation in gene expression levels (16-19). Performing new functional experiments was beyond the scope of this paper.

3. Nevertheless, in its current form, the manuscript is somewhat confusing and not easily to follow, and misses tables 2-7 and all supplementary tables.

Each supplementary table is presented as a separate worksheet in the same Excel file.

4. It is also not easy to pair figure legends to the figures.

We apologise for that and have rectified as appropriate.

5. How exactly were the 212 risk variants identified? How do we know GCTA?

The 212 variants were identified based on an approximation to conditional association analysis developed by Yang, Visscher and colleagues. This approximation does not require individual-level data for the studies included in the GWAS, only summary statistics; it is implemented in the widely used GCTA software package and is described in detail in the reference cited (PMID 22426310).

Briefly, approximate conditional association analysis involves the following steps, which are applied to each chromosome separately: (1) the variant with smallest p-value among all those with a $P < 3 \times 10^{-8}$ is identified – we call this a "sentinel risk variant"; (2) the association with every other variant within 10 Mb of that sentinel SNP is re-estimated after taking into account the association with that (and previously selected) sentinel SNP and the LD with it; and (3) steps 1 and 2 are repeated until no further variants are identified that have an association P-value $< 3 \times 10^{-8}$.

6. What eQTL datasets from tissues other than breast cancer were incorporated into the analysis? Are they sufficiently powered? From reading the manuscript, only the analysis of eQTL dataset related to whole blood is described sufficiently, and seem to have generated useful data.

These are all described, with their samples sizes and references, in Supplementary Table 2.

7. Are there significant associations of the 212 loci (SNPs) with gene expression in tissues other than breast that are not found in breast tissue – indicating that genetically-determined expression in immune cells or adipocyte is the driver of breast cancer risk? Having these associations in both breast and the other tissues may not be as informative to support a driver role of genetically-determined expression in immune cells or adipocyte.

Yes, this is shown in Tables 1 and 5, and described in the Discussion "(All but one of the novel target predictions were identified from eQTL analyses in blood, spleen or immune cells. Of note,

one novel target was identified through eQTL analyses in adipose tissue: ZNF703.)

8. Author identified 212 variants associated with breast cancer using the GCTA GWAS (supplementary table 1, which the reviewer could not find). As I understand this GWAS composes of the “original GWAS” (14910 cases and 17588 controls), iCOGS, and the “OncoArray” (61282 cases and 45494 controls) datasets. Not clear. Do all 212 SNPs have the same direction of risk in all three studies?

This question relates to previously published work, specifically the Michailidou et al. GWAS meta-analysis (PMID 29059683). We have amended the Methods as follows: “[Recently, Michailidou et al. (2) reported a breast cancer GWAS meta-analysis that] combined results from 13 studies: the OncoArray study (61,282 cases and 45,494 controls); the iCOGS study (46,785 cases and 42,892 controls); and 11 other individual GWAS (with a combined 14,910 cases and 17,588 controls). That is, a total of 122,977 cases and 105,974 controls.”

To specifically address the reviewer’s query about direction of effect, we have added a new column to Supplementary Table 1 with the direction of effect for all 212 sentinel SNPs, in each of the 13 studies.

In that new column, the 12th symbol is the direction for the iCOGS study, and the 13th symbol for the OncoArray study, the two single contributions with largest sample sizes. For all 212 sentinels, the direction of effect was the same in both of these studies (103 “- -” and 109 “+ +”).

9. Do the 20 SNPs in the joint analysis contain flipped alleles?

Only three of the 20 SNPs are A/T or C/G polymorphisms, which therefore could potentially be affected by cryptic strand differences between the Michailidou et al. GWAS and the UK Biobank data used in the GCTA joint association analysis. Of these 20 SNPs, only one had alleles flipped between the two datasets (rs11652463 on chr 17). However, this is NOT a problem, because one of the first steps in the joint analysis implemented in GCTA is precisely to compare the allele labels and frequencies between the GWAS results and LD reference panel: any strand inconsistencies are corrected (i.e. alleles are flipped in the reference dataset so that the labels and/or frequency matches that observed in the GWAS) prior to the analysis. That was the case for rs11652463, and so the conditional association results for this SNP are correct.

10. For sentinel 46 eQTL SNPs, author set the 1Mb gene boundary and r^2 to less than 0.05, and use a conservative p value 2.3×10^{-9} , however, for EUGENE the setting was r^2 0.1 and window size of 2Mb. When they apply settings for EUGENE, should the authors not use the same parameters? Otherwise, the authors need to discuss the effect of those parameters changes on the final results.

Yes, we agree that this difference in r^2 thresholds between the two analyses (prediction of target genes of sentinel risk variants vs. EUGENE gene-based analysis) causes unnecessary confusion.

We originally used a more liberal r^2 threshold (0.1) in the EUGENE analysis because EUGENE accounts for LD between sentinel eQTLs when calculating the gene-based P-value, and so we do not need to be as conservative as when using single eQTLs to predict target genes of sentinel risk variants.

Regarding the window size, there was a typo in the Methods section for the EUGENE analysis: we used a 1 Mb (not 2 Mb) window to the left and right of each gene, as in the target gene prediction analysis.

For consistency and to avoid confusion, we have repeated the EUGENE analysis with an r^2 threshold of 0.05 and now present those results instead of the original findings. Results are very similar between the two analyses (shown for overall breast cancer on the left panel in Figure below), although some of the original gene-based associations drop below the genome-wide significance threshold with the stricter r^2 threshold, which is consistent with reduced power (i.e. disease-associated eQTL no longer included in the gene-based test).

For completeness, we have also updated our eQTL database to include 31 new eQTL datasets (from 9 studies) and results from release V7 of GTex, which has a larger (up to 1.7-fold) sample size than release V6 included in the original submission. We also used a slightly more conservative P-value threshold to identify sentinel eQTL: 8.9×10^{-10} (which corrects for 55,765 transcripts, each tested for association with 1000 independent SNPs) instead of 2.3×10^{-9} (corrects for 21,742 protein coding genes, each tested against 1000 SNPs). The latter threshold did not account for non-coding RNAs tested in eQTL studies, which is now routine with RNA-seq data. As shown on the right panel above, gene-based association results were very similar between the original and the updated eQTL databases.

Overall, with these analytical changes, we now report the identification of 6 novel loci for overall breast cancer and 4 for ER-negative (instead of 9 and 4, respectively, in the original submission). We have also made available as Supplementary Material the EUGENE association results for all genes tested.

The list of predicted target genes of sentinel risk variants was also updated: we found a similar number of targets (88 for overall and 24 for ER-negative breast cancer), most of which (92% and 88%, respectively) were identified as targets based on both the original and updated eQTL databases.

11. Also for the sentinel eQTL SNPs, are those low frequency SNPs, what is the MAF distribution of them? Should a setting for effect size be applied? What would be the implication of low frequency SNPs?

The MAF of each sentinel eQTL included in the EUGENE analysis has been added to Supplementary Tables 6 (overall breast cancer analysis) and new Supplementary Table 12 (ER-negative breast cancer).

For the 11 genes identified in the EUGENE analysis of the overall breast cancer GWAS, which had a large sample size ($N=228,951$), 14 (21%) of the 66 sentinel eQTLs were relatively uncommon, with a MAF between 1% and 5%. All others had a $MAF > 5\%$.

In the ER-negative analysis ($N=140,970$), all 10 sentinel eQTLs for the 4 genes identified by EUGENE had a $MAF > 5\%$.

We applied a filter for the eQTL association P-value (specifically a $P < 8.9 \times 10^{-10}$), which is determined not just by the eQTL effect size (i.e. beta) but, importantly, also its standard error (SE). This is essential to (1) exclude from the EUGENE analysis eQTL identified with low confidence (i.e. high SE); and (2) apply a correction for multiple gene and SNP testing when identifying significant eQTL from published studies.

Most sentinel eQTLs were common (i.e. $MAF > 5\%$) because most eQTL studies published have a relatively small sample size and so typically have restricted their analysis to common SNPs. As eQTL studies and GWAS become larger, we expect that many more low frequency eQTLs are identified and found to have an association with breast cancer risk.

12. Authors queried 86 eQTL datasets (supplementary table 2, could not see it), what is the sample size for the different tissues (adipose, breast, immune cells, spleen and whole blood (blood monocytes?)), are they very different, and the whole blood datasets are the major source?

Yes, the whole blood datasets have the largest sample size, as described in Supplementary Table 2, and alluded to in the Discussion (“However, another possibility is that eQTL detected in well-powered studies of blood are predictors of eQTL in other less accessible tissues, including breast and adipose tissue.”)

13. What is whole blood? PBMCs?

The term “whole blood” is used in eQTL studies that extracted RNA from all cell types present in peripheral blood, include PBMCs, granulocytes and red blood cells.

14. For the direction effect of gene expression on cancer risk, should author consider the known oncogenic or tumor suppressor feature of the genes? The allele effect on gene expression should be consistent with known oncogene or tumor suppressor function, or should be discussed if different.

We were able to assign direction effect of gene expression on cancer risk to 78 candidate target genes. Of these 78 genes, only very few have been unequivocally classified as tumor suppressor or oncogenes in breast tumours: for example, ZNF703 is a well known oncogene in breast cancer and decreased expression is associated with decreased risk. Similarly, oncogenic activity has been reported for PIK3C2B (PMID 26934321), for which we found that decreased expression is associated with decreased risk. Another gene in which decreased expression is associated with decreased risk is PTPN22 which is known to negatively regulate antigen presentation and therefore might suppress immunoelimination. In contrast, CCNE1 (PMID 19107593) and APOBEC3A (PMID 16720547) have been reported to have an oncogenic roles, but we found that increased expression was associated with decreased risk. Interestingly, we have previously found the same counterintuitive relationship between breast cancer risk alleles and CCND1 expression (PMID 23540573).

In order to evaluate this issue more systematically, we compared the expression of the candidate target risk genes in adjacent 'normal' tissue vs breast tumours in TCGA. Overall the direction of expression was consistent for 62.5% of genes associated with ER-ve breast cancer risk (genes for which we found that decreased expression is associated with decreased risk are noted in red in the first column, and those for which increased expression was associated with decreased risk in blue).

However, for the candidate target genes we identified as being associated with overall breast cancer risk the direction of expression was consistent in TCGA for 42.5% of genes.

There are several possible explanations for these apparent inconsistencies in direction. For example, the 'normal' tissue profiled in TCGA is unlikely to accurately reflect expression levels in the progenitor cells that give rise to breast tumours. In addition, many genes have pleiotropic functions and show tissue context-dependent oncogenic or suppressor roles (eg PMID 23851498 and 21948802). Similarly, the TGF- β signalling pathway has a dual role, acting both in initial tumour development and in later tumour progression (PMID 22992590). Another explanation may be that many of our predicted target genes (e.g. ZBTB38, RHBDD3) do not modulate disease risk through breast tissue but instead act in other tissues, for example, immune cells.

For this reason, we have not included the TCGA data in the manuscript but we have added the following information on some known oncogenes/tumor suppressor genes:

In some cases, this was consistent with their known function. For example, ZNF703 is a well known oncogene in breast cancer (29) and decreased expression is associated with decreased risk. Similarly, oncogenic activity has been reported for PIK2C2B (46), for which we found that decreased expression is associated with decreased risk. Another gene in which decreased expression is associated with decreased risk is PTPN22 which is known to negatively regulate antigen presentation (47) and

therefore might suppress immunoelimination. In contrast, *CCNE1* (48) and *APOBEC3A* (49) have been reported to have an oncogenic roles, but we found that increased expression was associated with decreased risk. Interestingly, we have previously found the same counterintuitive relationship between breast cancer risk alleles and *CCND1* expression (50). However, the expression patterns observed in breast tumors may not be relevant to the activity of these genes in the progenitor cells that give rise to breast tumors.

15. Author applied S-PrediXscan using 922 whole blood samples from the Battle dataset. This dataset came from a study of depression with 463 subjects with depression and 459 subjects without it. Is there a concern of selection bias?

To specifically address this concern, we repeated the S-PrediXscan analysis using GTEx whole-blood (n=369) as the reference transcriptome dataset. Results (presented in Supplementary Table 6) are highly concordant with those obtained with the larger DGN dataset.

16. Why did the authors not apply EUGENE to each tissue? Wouldn't tissue-specific data be informative?

How informative tissue-specific data might be depends on the goal of the analysis. On the one hand, if the goal is to identify likely target genes of sentinel risk variants, then tissue-specific data would more informative. This is exactly what we did to identify likely target genes of the 212 sentinel risk variants.

On the other hand, if the goal is to identify novel loci with variants associated with breast cancer risk using gene-based analysis, then it is more powerful to consider in a single analysis eQTL identified when considering information from multiple tissues. This approach is more powerful than considering each tissue at a time for two main reasons. First, because tissue-specific eQTL are common, and so a multi-tissue analysis is able to capture the association between these and breast cancer risk in a single test. Second, because in single-tissue analyses, one needs to appropriately account for testing multiple tissues, thereby decreasing the P-value threshold for experiment-wide significance, which decreases power.

This information has been added to the Discussion (page 22).

17. Gene-based tests and eQTL analysis led to similar findings, correct?

Correct (lines 534 to 536).

18. Page 17, Oncoscore analysis. The authors analyzed 47 genes. Why 47 genes?

We can now analyze 100/114 genes using an updated version of OncoGene. The remainder (which are all pseudogenes, or non-coding genes) are not evaluated by OncoScore.

19. Page 22, targeting *KCNN4*. What is the magnitude of association of the genetically-determined gene expression with breast cancer risk?

*The 71 SNPs used by S-PrediXscan to predict genetically-determined *KCNN4* expression explained ~50% of the variation in gene expression measured in whole blood (as indicated in Supplementary Table 5). In turn, the association between genetically-determined *KCNN4* expression and breast cancer had a magnitude of 0.0538 (beta), with a SE of 0.0065. This corresponds to a Z-score of 8.2 and a P-value of 2×10^{-16} . The Z-score and P-value were included in Supplementary Table 5. We have now updated this table to also include the corresponding beta and SE.*

20. Would it make sense to develop a drug that targets *KCNN4* for disease prevention? All drugs have also side effects.

*We have edited the discussion as follows: "For example, decreased risk of ER-negative breast cancer was associated with decreased expression of *KCNN4*, suggesting that an antagonist that targets this potassium channel and has a good safely profile (42) might reduce disease risk. Given these results,*

we suggest that KCNN4 should be prioritised for functional and pre-clinical follow up”).

Reviewer #2 (Remarks to the Author):

Authors would strengthen the publication by including a more robust discussion of why they believe the tools they chose are the most appropriate and thus reveal biologically accurate results.

We have added the sentence below to the Discussion to address this remark. Please also response to question 16 of Reviewer #1.

“[We also identified novel breast cancer risk loci using the recently-described EUGENE gene-based association test (6, 7)], which was developed to aggregate evidence for association with a disease or trait across multiple SNPs. Unlike other similar gene-based methods (e.g. S-PrediXcan), EUGENE includes in a single test information from eQTL identified in multiple tissues; this property is expected to increase power to detect gene associations when multiple cell types/tissues contribute to disease pathophysiology, for two main reasons. First, because tissue-specific eQTL are common, and so a multi-tissue analysis is able to capture the association between all known eQTL and disease risk in a single test. Second, because in single-tissue analyses, one needs to appropriately account for testing multiple tissues, thereby decreasing the significance threshold required for experiment-wide significance, which decreases power.”

Reviewer #3 (Remarks to the Author):

1. Main Table 1 presents, “Novel target gene predictions of sentinel risk variants for overall breast cancer.” It appears that at known GWAS-identified risk loci for overall breast cancer, the same sentinel risk SNPs or SNPs in strong LD with these SNPs are often predicted to target more than one gene. For example, when the first row of main Table 1 is read in conjunction with Supplementary Table 4 and the corresponding regional association plot, at the 2p23.3 locus alone, there are four predicted targets – ADCY3, CENPO, DNAJC27, and RP11-443B20.1.

-- How is this to be interpreted in the absence of experimental evidence?

-- Are the effects of the sentinel risk SNPs on breast cancer mediated by the expression of all four genes – is such multiplicity of genetic regulation plausible? Is this simply an artefact due to nearby genes being co-expressed?

-- Should one use the OncoScore to prioritise CENPO, which has the higher OncoScore, over the other three genes?

-- Do all four eQTL signals colocalise with the sentinel risk SNPs or can colocalisation be used to further dissect this and other similar loci?

We were able to predict at least one target gene for 46 of the 212 sentinel risk variants for overall breast cancer. Of these 46 SNPs, 25 had a single target gene predicted, 10 had two, and 11, including the rs6725517 variant in ADCY3 that the reviewer refers to, had three or more (maximum of 6). All target gene predictions were listed in Supplementary Table 3, and are now summarised in a more intuitive fashion in the new Supplementary Tables 4 (for overall breast cancer) and 12 (for ER-negative breast cancer).

One likely explanation for multiple target genes being predicted for a single sentinel risk variant is that it tags a single functional variant in a regulatory element that controls the expression of multiple genes. There are many examples of this in the literature, for example: Sanyal et al. (Nature 2012, PMID:22955621), Burren et al. (Genome Biol 2017, PMID:28870212), and Javierre et al. (Cell 2016 PMID:27863249). Another possible explanation is that the sentinel risk variant tags multiple

functional variants (not all relevant to breast cancer) which overlap distinct regulatory elements that control the expression of different genes. How can we distinguish between these two biologically-distinct scenarios without extensive functional follow up experiments?

Various statistical methods have been developed to distinguish coincidental (i.e. the sentinel risk SNP and the sentinel eQTL tag different underlying functional variants) from shared (i.e. the sentinel risk SNP and the sentinel eQTL tag the same underlying functional variant) associations, namely RTC, coloc, eCAVIAR, HEIDI/SMR and JLIM. However, as Chun et al. recently showed (Nat Genet 2017, PMID 28218759), the leading methods are of limited use at r^2 levels above 0.8, the threshold we used to identify sentinel eQTL in LD with sentinel risk SNPs. This is, at high r^2 levels, these methods have an extremely high false-positive rate (see Supp Table 3 in their paper). For this reason, we do not advocate using such methods to distinguish between coincidental or shared effects when the sentinel eQTL and sentinel risk variants have an $r^2 > 0.8$.

Instead, to help prioritise target genes for experimental validation, we now report in the new Supplementary Tables 8 and 17 a list of genes that are predicted to be the targets of sentinel risk variants based not just on eQTL information, but also on the presence of:

- (1) Chromatin interactions between enhancers that contain a sentinel risk SNP (or a proxy with $r^2 > 0.95$) and the respective gene promoter. Chromatin interactions were identified based on ChIA-PET (PMID 22265404), capture Hi-C (PMID 25938943 and 27863249) and in situ Hi-C (PMID 25497547); or*
- (2) Correlations between epigenetic marks of enhancers that contain a sentinel risk SNP (or a proxy with $r^2 > 0.95$) and gene expression levels. These correlations were reported by PreSTIGE (PMID 24196873), FANTOM5 (PMID 24670763), Hnisz et al. (PMID 24119843) and IM-PET (PMID 24821768).*

We considered data from all blood cell types analysed in these studies, given that most target genes were identified based on eQTL data from whole-blood.

Of the 88 predicted target genes for overall breast cancer, 25 were supported by evidence from (1) or (2), while for ER-negative target genes functional support was observed for 18 out of the 24 genes. For example, of the three genes that were predicted to be the targets of rs6725517 based on eQTL data, two (CENPO and ADCY3) are supported by the functional data listed above.

This additional information can be used to prioritise target gene predictions for experimental validation, which we now stress in the Discussion, which concludes that 'Further investigation into the function of the genes identified in breast and immune cells, particularly those which have additional support from experimental or computational predictions of chromatin looping, should provide additional insight into the etiology of breast cancer.'

2. Further, there is strong evidence for an eQTL in TCGA breast cancer tissue for ADCY3 that colocalises with the sentinel/credible causal breast cancer risk SNPs at this locus (Supplementary Table 18 in Michailidou et al. Nature 2017; column P: risk esnp p/best esnp p = 28). Isn't a tissue-specific eQTL that colocalises more likely to represent the true target gene given that there is also evidence favouring ADCY3 from the authors' present analysis and another recent breast tissue-specific eQTL analysis of breast cancer risk loci (Guo et al. AJHG 2018)?

Likewise, there is strong eQTL and colocalisation evidence for AMFR at the 16q12.2 locus in normal breast tissues from the authors' 2017 paper and the current analysis and it is quite possible that the novel gene highlighted by the authors at this locus (CES1) is a false positive given that it also has a lower OncoScore than AMFR. Along the same lines, it is very difficult to choose between STXBP4 and HLF with the breast-specific evidence pointing to the former. All the above limitations should be highlighted and discussed. Far more caution is urged in the presentation of these results as novel findings implicating immunological regulation at breast cancer risk loci rather than the result of

routine co-expression of neighbouring genes in eukaryotic genomes being picked up by the use of dozens of eQTL data sets all together (see for example Michalak, Genomics 2008).

Given the limitations of OncoScore (being based only on text mining) and the context-dependency and tissue-specificity of eQTLs, we do not necessarily think that ADCY3 is the most likely target at this locus. While the TCGA data are important, the most appropriate tissue might, for example, be adolescent breast tissue stimulated by estrogen, or luminal progenitor cells from normal breast tissue, but data from such tissues are not available. For this reason we do not claim to have found the target genes, but candidate target genes, that need to be confirmed by experimental studies. Similarly we would not prioritise AMFR over CES1, or STXBP4 over HLF, based entirely on the OncoScore. However, ChIA-PET and capture Hi-C evidence provide additional support for ADCY3, CENPO, AMFR and HLF as target genes. Overall there was additional functional support for 39 of 112 candidate target genes, which should be prioritised for functional follow up.

3. It may be helpful for the reader to weigh the novelty of the eQTL evidence for themselves if all loci in Table 1 where the novel target gene prediction is only one of multiple (please specify the number) target genes predicted at the same locus are flagged in a separate column. A truly “novel” gene in the context of the authors’ aims (Introduction, pages 11 and 12) might be one where there is no reasonable breast tissue-specific eQTL prediction but where the authors’ multi-tissue eQTL prediction has revealed a target gene. Please highlight in Table 1 the target genes with breast tissue-specific eQTL evidence (if any) at such loci. The same comments also apply to Table 5.

Done, as requested. Tables 1 and 5 now include association results between eQTL and gene expression in breast tissue (based on GTEx V7). We have also added results for GTEx blood for comparison.

4. Why was the EUGENE method not applied to the known breast cancer risk loci? It would be interesting to compare the EUGENE predictions for the known breast cancer risk loci to the authors’ eQTL-based predictions for the same loci. While I accept the authors’ rationale for applying EUGENE to adjusted GWAS summary statistics for the identification of completely new breast cancer risk loci (Methods, page 27), this does not preclude the application of EUGENE to the original GWAS summary statistics at the known loci for comparison against eQTL findings.

We did not originally apply EUGENE to known breast cancer risk loci because EUGENE was not developed to identify likely target genes of individual sentinel risk variants. Instead, it was designed to combine information across multiple eQTL to identify novel risk loci.

There are two ways in which we could learn from applying EUGENE to known breast cancer risk loci and comparing results with target predictions from the single eQTL analysis.

First, we could apply EUGENE to the original GWAS results to identify significant gene-based associations for genes that were not predicted as targets by the single eQTL analysis. However, such associations could be challenging to interpret. For example, if a sentinel risk SNP has an extremely significant association with breast cancer risk in the original GWAS (e.g. rs2981579, $P=10^{-316}$), then a nearby sentinel eQTL for gene X that is in low LD (e.g. $r^2=0.05$) with that sentinel risk variant might still have a very significant association with breast cancer risk (e.g. a 95% attenuation of the association chi-square for rs2981579 would yield a $P=10^{-16}$). In this scenario, the sentinel eQTL would not be in high LD with the sentinel risk variant (and so gene X would not be predicted as a target based on the individual eQTL analysis), but gene X would have a very significant gene-based association. The latter would not be particularly interesting, as it could simply represent a ‘passenger’ association. Because of this potential difficulty in interpreting such associations, we don’t think EUGENE (or other gene-based tests, for that matter) should be applied to the original GWAS findings to identify additional targets amongst known breast cancer risk loci.

Second, we could ask if the targets of sentinel risk variants, identified based on individual eQTL, also had a significant EUGENE association in the adjusted GWAS results (ie. adjusted for the effects of the

212 sentinel variants). If so, then it would indicate that additional breast cancer risk variants (i.e. that are not genome-wide significant) supported our original target gene prediction. This additional information could potentially help prioritize genes for functional follow-up.

Given the arguments above, and the reviewer's interest, we applied EUGENE to the predicted target genes of sentinel risk variants, using the adjusted (but not the original) GWAS results.

Of the 88 genes predicted as targets of overall breast cancer sentinel risk variants, 11 had a nominally significant gene-based association ($P < 0.05$), but only one remained significant after correcting for multiple testing ($P < 0.05/88 = 0.00057$): CBX6 ($P = 0.0002$). For ER-negative breast cancer, four of the 24 target genes had a nominally significant association, with only one significant after correcting for multiple testing ($P < 0.05/24 = 0.002$): RALB ($P = 0.0002$).

5. Since OncoScore uses text-mining of the published biomedical literature to link gene names with cancer, would it not be contaminated by the fact that several of these genes already find mention in breast cancer GWAS papers due to their proximity to the reported risk loci for this cancer? This could lead to spuriously elevated scores for some genes and its influence on the OncoScore is not clear to me.

It is possible that the GWAS literature could contribute slightly to the OncoScore for known risk loci, although we used a OncoScore cut-off threshold of 21.09 for novelty, so we consider that our conclusion that "44/100 scored below the recommended OncoScore cut-off threshold (21.09) for novelty, including 25 with an OncoScore of 0, indicating no prior evidence for a role in cancer biology" is appropriately conservative. Of the 66 genes with an OncoScore >21.09, only 15 have been previously mentioned in breast cancer GWAS papers, and for all these (MDM4, FAM175, CASP8, CCDC170, NPAT, PRC1, ESR1, TGFBR2, ATM, L3MBTL3, GATAD2A, PIDD1, ATG10, MAN2C1 and AKAP9) there is extensive literature concerning their role in cancer in addition to their mention in these papers.

6. Would a look-up of gene essentiality in cancer or immune cell lines provide a more unbiased measure of carcinogenic/immune regulatory potential?

This an interesting suggestion but a) gene 'essentiality' is a very limited measure of all the hallmarks of cancer, and b) genome-wide studies of essential genes are not available on all the tissues we queried in this study. The papers (eg PMID 29033457) are based on cell lines which are many not reflect well essentiality in primary cells, given the context dependency of 'essentiality'.

7. The question about multiple target genes in the same putative breast cancer susceptibility region also arises for the EUGENE results in Table 3, in particular for the three GSTMx transcripts at 1p13.3 that are simply likely to be co-expressed together. The ultimate focus of EUGENE on identifying sub-GWAS-significant risk loci rather than the current emphasis on the biological function of specific genes should perhaps be reiterated in the discussion.

The focus of EUGENE has now been stressed in the discussion (page 22).

8. Discussion, page 19: "However, another possibility is that eQTLs detected in well-powered studies of blood are predictors of eQTL in other less accessible tissues, including breast and adipose tissue." Since the authors use 183 GTEx normal breast tissue samples in their analysis could this possibility be formally tested? It may be helpful to those broadly interested in the functional follow-up of GWAS to know how frequently do eQTLs identified in the largest blood eQTL data set analysed by the authors also show up at less significant p-value thresholds in the GTEx normal breast tissues.

This is an important question that has been thoroughly addressed in a recent study by Yang and colleagues (PMID 29891976), which we now cite. They found that genetic effects are highly correlated between blood and brain for top eQTLs.

In line with this, we also found that the sentinel eQTL used to identify target genes for overall breast cancer had an effect on gene expression that was broadly consistent between blood and breast tissue, based on data from GTEx V7, although there were a few notable exceptions (see Figure below, which shows 70 of 88 eQTL with available results in both tissues).

The additional data we have added to Tables 1 and 5 (eQTL association in GTEx V7 breast and blood) can be used to explicitly determine which of the target genes would also be identified based on a smaller breast eQTL dataset ($n=251$) and a more liberal eQTL P -value threshold. For example, of the 88 eQTL listed in Table 1, 74 were tested for association in the GTEx breast eQTL dataset, of which 33 (45%) had a $P < 0.05$. For most of these (80%), the directional effect of the eQTL on gene expression was the same between breast and whole-blood.

REVIEWERS' COMMENTS:

Reviewer #1 (Remarks to the Author):

The authors adequately addressed the raised questions and concerns. No further questions.

Reviewer #3 (Remarks to the Author):

The authors have adequately addressed all my questions. The inclusion of epigenetic and chromatin interaction data for gene prioritisation, the comparison between GTEx breast and blood eQTLs, and all the revised tables are particularly helpful.